# Hormetic Nutrition and Redox Regulation in Gut–Brain Axis Disorders

**DOI:** 10.3390/antiox13040484

**Published:** 2024-04-18

**Authors:** Maria Scuto, Francesco Rampulla, Giuseppe Maria Reali, Sestina Maria Spanò, Angela Trovato Salinaro, Vittorio Calabrese

**Affiliations:** Department of Biomedical and Biotechnological Sciences, University of Catania, 95124 Catania, Italy; francescorampulla1985@virgilio.it (F.R.); giuseppe.reali@gmail.com (G.M.R.); sestinamaria.spano@gmail.com (S.M.S.); calabres@unict.it (V.C.)

**Keywords:** Gut–brain axis, inflammation, Nrf2 pathway, hormesis, probiotics, polyphenols, neurological disorders, organoids

## Abstract

The antioxidant and anti-inflammatory effects of hormetic nutrition for enhancing stress resilience and overall human health have received much attention. Recently, the gut–brain axis has attracted prominent interest for preventing and therapeutically impacting neuropathologies and gastrointestinal diseases. Polyphenols and polyphenol-combined nanoparticles in synergy with probiotics have shown to improve gut bioavailability and blood–brain barrier (BBB) permeability, thus inhibiting the oxidative stress, metabolic dysfunction and inflammation linked to gut dysbiosis and ultimately the onset and progression of central nervous system (CNS) disorders. In accordance with hormesis, polyphenols display biphasic dose–response effects by activating at a low dose the Nrf2 pathway resulting in the upregulation of antioxidant *vitagenes*, as in the case of heme oxygenase-1 upregulated by hidrox^®^ or curcumin and sirtuin-1 activated by resveratrol to inhibit reactive oxygen species (ROS) overproduction, microbiota dysfunction and neurotoxic damage. Importantly, modulation of the composition and function of the gut microbiota through polyphenols and/or probiotics enhances the abundance of beneficial bacteria and can prevent and treat Alzheimer’s disease and other neurological disorders. Interestingly, dysregulation of the Nrf2 pathway in the gut and the brain can exacerbate selective susceptibility under neuroinflammatory conditions to CNS disorders due to the high vulnerability of vagal sensory neurons to oxidative stress. Herein, we aimed to discuss hormetic nutrients, including polyphenols and/or probiotics, targeting the Nrf2 pathway and *vitagenes* for the development of promising neuroprotective and therapeutic strategies to suppress oxidative stress, inflammation and microbiota deregulation, and consequently improve cognitive performance and brain health. In this review, we also explore interactions of the gut–brain axis based on sophisticated and cutting-edge technologies for novel anti-neuroinflammatory approaches and personalized nutritional therapies.

## 1. Introduction

Growing evidence has highlighted the interaction of the gut–brain axis as a critical determinant of human health and/or disease and a key regulator of host physiology, inflammatory responses and redox homeostasis [1]. The high prevalence of gut and brain disorders poses serious public health challenges worldwide [2,3]. Perturbations in gut microbiota composition are characterized by increased oxidative stress, neuroinflammation and progressive neuronal death, leading to the development of brain disorders, particularly Alzheimer’s disease (AD), Parkinson’s disease (PD), schizophrenia, autism spectrum disorder (ASD), depression and anxiety [4,5,6,7,8]. For this reason, the gut microbiota, the complex ecosystem of microorganisms in the gastrointestinal tract, is also called “the second brain” or “the forgotten endocrine organ” [9]. Indeed, an imbalance of the gut microbiota can lead to increased permeability of the intestinal epithelial barrier (IEB) with the release of proinflammatory cytokines and promotion of a neuroinflammatory response [10]. Recently, preclinical evidence suggests that gut dysbiosis is closely associated with increased circulating neurotoxic mediators, such as TNF-α, IL-1β, IL-6 and C-reactive protein, which can directly cause low-grade inflammation predisposing to progressive gastrointestinal and nervous system disorders [11,12]. Furthermore, patients with brain damage and amyloidosis exhibit a perturbated gut microbiome with the abundance of *Escherichia*/*Shigella* significantly associated with an increase in proinflammatory cytokine gene profiles and a reduction in the abundance in the anti-inflammatory *Eubacterium rectale* [13]. It is well established that oxidative stress induces progressive degeneration and neuronal death [14]. Specifically, the imbalance between free-radical production and the efficiency of antioxidant defense pathways triggers neuronal dysfunction and degeneration, being vagal sensory neurons particularly vulnerable to reactive byproducts species of oxidative stress including hydrogen peroxide, 4-hydroxynonenal (4-HNE), malondialdehyde (MDA) and acrolein [15]. It is interesting to note that moderate stress is protective for the organism because it activates adaptive resilience pathways of crucial importance for maintaining cell survival and an optimal quality of life. However, when stress exceeds the intracellular defense capacity, sensory neurons to counteract ROS overproduction activate neuroprotective pathways involving the translocation of antioxidant nuclear factor erythroid 2–related factor 2 (Nrf2) from cytosol into the nucleus and the upregulation of detoxification phase II enzymes, termed *vitagenes*, particularly heat shock protein 70 (Hsp70), heme oxygenase-1 (HO-1), sirtuin-1 (Sirt1), thioredoxin (Trx)/thioredoxin reductase system, NADPH quinone oxidoreductase 1 (NQO1) and γ-glutamylcysteine synthetase (γ-GCs), which is part of the glutathione redox system to protect against the initiation and progression of neurodegenerative disorders [16,17,18,19,20,21,22]. Likewise, the disequilibrium in the function and composition of the gut microbiota promotes the production of toxic metabolites and proinflammatory cytokines that destroy the IEB with the activation of local and distant immune cells and deregulation of enteric neurons, astrocytes and microglial cells, which ultimately reduces the amount of beneficial substances such as short-chain fatty acids (SCFAs), neurotransmitters such as gamma amino butyric acid (GABA), acetylcholine, serotonin, norepinephrine, histamine and anti-inflammatory lipid mediators such as lipoxin A4 [23,24]. This culminates in a BBB dysfunction that triggers a vicious circle converging in neuroinflammation, and predisposing neuronal and glial cells to apoptotic death, primarily in the cerebral cortex and hippocampus, underlying the development of dementia [25]. Of note, gut dysbiosis associated to the onset and progression of neurodegeneration is characterized by a significative reduction in specific microbial species belonging to the *Firmicutes* and *Bifidobacterium* phyla, and an increase in pathogenic species, mainly pro-inflammatory bacteria of the *Proteobacteria* and *Bacteroidetes* phyla [26]. Similarly, individuals with neuropsychiatric disorders (i.e., depression, anxiety, schizophrenia and ASD) showed higher levels of *Enterobacteriaceae*, *Alistipes* and *Clostridium*, and lower levels of *Faecalibacterium* [27,28,29,30]. Therefore, the gut–brain axis is emerging as an attractive target for the discovery of new natural dietary supplements to restore intestinal inflammation due to alterations in gut microbiota bacteria for brain health and cognitive function [31,32]. The human gut metabolome is the result of a complex chemical interplay between commensal microbes and their hosts. Indeed, the gut microbiota produces thousands of metabolites which specifically interact with various bacterial species and strains, dietary interventions (e.g., probiotics and polyphenols) and host molecules, including amyloids and dopamine, which may reach the brain to regulate neurological function. In particular, the gut microbiota produces SCFAs that strengthen the intestinal barrier, stimulate the intestinal L-cell to secrete glucagon-like peptide 1 (GLP-1) and gastrointestinal peptide (GIP) to directly or indirectly inhibit NLR family pyrin domain-containing 3 (NLRP3) inflammasome activation, insulin resistance and ROS [33]. Furthermore, SCFAs stimulate intestinal gluconeogenesis (IGN) in intestinal epithelial cells and, together with GLP-1/GIP, activate the vagus nerve from the enteric nervous system (ENS) and brain-derived neurotrophic factor (BDNF) signaling in the brain, reinforcing the importance of gut and neural signaling networks to promote brain health [33]. Importantly, a meta-analysis study observed that dietary anthocyanins effectively reduced the *Firmicutes*/*Bacteroidetes* ratio and increased the content of SCFAs including acetic acid, propionic acid and butyric acid to improve gut health depending on the duration and dosage of treatment [34]. Recent clinical evidence reported that SCFAs levels are altered in the fecal and urine samples of patients with ASD [35], as well as in gut and feces among patients with AD and PD [36]. The interplay between polyphenols and gut–brain axis signaling provides an important tool to modulate gut microbiota composition and the abundance of specific beneficial bacteria for maintaining intestinal barrier integrity and CNS function [37]. Furthermore, the dietary intake of probiotics in adequate doses interferes with potential pathogens alleviating the symptoms not only of inflammatory bowel diseases (IBD) but also of neuropsychiatric and degenerative disorders via the gut–brain axis in experimental models [38] and in patients [39,40]. This review integrates the emerging literature focused on the molecular mechanisms and cellular pathways linking gut microbiota disorders to IEB and BBB dysfunctions, underlying the development of nervous system disorders. Therefore, identifying the mechanisms by which specific vital gut microorganisms and/or dietary natural compounds influence intestinal immune cells or potentially cross the BBB to affect central nervous system cells will help us achieve a deeper comprehension of gut–brain axis crosstalk for human health. Finally, antioxidant and anti-inflammatory signaling pathways activated by hormetic nutrients including polyphenols alone and/or in synergy with probiotics using the innovative organoid technology for microbiota-based neuroprotective and anti-neuroinflammatory therapeutic options are also discussed.

## 2. Hormetic Nutrition and Redox Resilience Signaling in Gut–Brain Axis

Hormetic nutrition is a novel idea that emerged from the extensive research over the past two decades on the occurrence of hormesis in the biological and biomedical sciences, especially in the areas of food science and nutrition. Hormesis represents the confluence of a wide range of adaptive strategies that mediate the upregulation of several highly co-ordinated series of processes at the cell, organ and organismic levels to protect against endogenous and exogenous stress insults and/or events. The hormesis adaptions are expressed via the occurrence of dose–time responses that are biphasic with a low or moderate dose stimulation and a higher dose inhibition. Interestingly, the low-dosage stimulating responses result in enhanced stress resilience/adaptive capacity via anti-inflammatory and antioxidant molecular networks that promote healthy aging and longevity, prevent the onset or slow the progression of neurodegenerative and chronic gastrointestinal diseases. Thus, the hormetic dose response integrates the responses of both single natural compounds and complex mixtures in their biological responses, showing consistency with the quantitative features of the hormetic pattern in the gut–brain axis. It is important to remember that toxic compounds including ROS, inflammatory cytokines and microbial metabolites are also essential at low levels in the initial phases, as they create minimal stress which is fundamental for the activation of antioxidant pathways and resilience mechanisms promoting gut barrier integrity and neuroprotection. Of relevance, the cellular stress resilience response activated by hormetic nutrients is emerging as a promising preventive and therapeutic strategy to counteract oxidative stress, inflammatory states and gut microbiota dysregulation occurring both in gastrointestinal and brain disorders [41,42]. Specifically, numerous polyphenols, including resveratrol, hidrox^®^, curcumin, *Crocus sativus* L., mushrooms and blueberry modulate and upregulate the Nrf2 signaling pathway and stress resilience *vitagenes* to preserve gut and brain homeostasis during pathological processes [43,44,45,46,47,48]. Several preclinical and clinical studies highlighted that hormetic nutrition with appropriate dosages of antioxidant polyphenols alone and/or in synergistic action with probiotics shows powerful neuroprotective and therapeutic effects by activating the Nrf2 pathway and related *vitagenes* to reduce gut dysbiosis and attenuate mitochondrial dysfunction, neuroinflammation and cognitive impairment, as well as symptoms of schizophrenia and depression (Figure 1) [49,50,51,52,53,54]. The neuroprotective potential of these natural substances is thought to be mediated via the ENS, neuroendocrine system and vagus nerve, as well as the spinal nerves of the gut [55,56,57,58,59,60,61]. In this scenario, probiotics also refer to “*psychobiotics*”, candidate species of live symbiotic bacteria that, when ingested in adequate amounts, confer brain health effects to the host for an innovative nutritional approach targeting gut microbiota to treat various neurological and psychiatric disorders [62,63]. Notably, probiotics, such as *Lactobacillus paracasei*, *Lactobacillus acidophilus*, *Lactobacillus casei*, *Lactobacillus fermentum*, *Lactobacillus helveticus*, *Lactobacillus rhamnosus*, *Bifidobacterium bifidum*, *Bifidobacterium longum*, *Bifidobacterium breve* and *Bifidobacterium infantis*, improve the central expression of BDNF, N-methyl-D-aspartatic acid (NMDA) receptor and other neuroactive peptides involved in synaptic and neural plasticity to enhance memory, cognition and behavior, and reduce microglial activation in a wide range of neurological illness including anxiety and depression [64,65,66,67].

A moderate intake of hormetic nutrients including polyphenols and psychobiotics promotes gut and brain healthy effects through the activation of Nrf2 signaling and stress resilience *vitagenes* [62,68,69]. In particular, polyphenols (i.e., hidrox^®^, curcumin, sulforaphane, resveratrol, blueberry and mushrooms), in synergy with psychobiotics, particularly the *Bifidobacterium* and *lactobacillus* phyla, upregulate antioxidant *vitagenes* including Hsp70, HO-1, γ-GCs, Sirt1 and Trx to inhibit oxidative stress and inflammatory NF-κB dependent pathways (i.e., TNF-α, IL-6 and IL-1β), leading to gut dysbiosis and the onset of CNS disorders via microbiota–vagus nerve–brain interactions. Therefore, hormetic nutrients can be considered promising therapeutic agents capable of restoring gut homeostasis and brain health during pathological conditions. 

The possibility of adopting natural antioxidant therapies in the prevention and initial phases of intestinal and neuronal disorders has been documented as being able to restore or delay the subsequent phases or to achieve remission [61,63]. Therefore, hormetic nutrition is a new evolving field in its application to gut and brain health, in both preventative and restorative study models. Accordingly, hormesis establishes that a low dose of polyphenols and/or polyphenol-combined nanoparticles, in synergy with probiotics, may modulate multiple antioxidant avenues, including Nrf2 and cellular phase II antioxidant enzymes (e.g., superoxide dismutase, catalase, glutathione peroxidases, thioredoxins/thioredoxin reductases, peroxiredoxins) and non-enzymatic antioxidant molecules (e.g., glutathione, hydrogen sulfide, carnosine, coenzyme Q and bilirubin), ultimately acting as neuroprotective and therapeutic agents [55]. In fact, the moderate consumption of natural compounds exerts gut and brain healthy effects by promoting the activation of stress resilience *vitagenes* and the inhibition of NF-κB-dependent pro-inflammatory pathways [53,56]. This postulates antioxidant *vitagenes* as a promising target for anti-neurodegenerative approaches based on the use of polyphenols in order to induce powerful neuroprotective and anti-neuroinflammatory effects in synergy with probiotics of clinical relevance, as a potential option to drugs, or the enhancement of classical neuro- and gastrointestinal therapies [58]. However, in the brain, a high dose of polyphenols can produce neurotoxicity related to the increased production of ROS and inflammatory markers with subsequent neuronal apoptosis or depletion of cellular resilience pathways [60]. The bidirectional crosstalk between the intestinal microflora and the brain occurs through a variety of pathways, including the vagus nerve, immune system, metabolic and neuroendocrine pathways and bacteria-derived metabolites. Compelling evidence suggests a critical role of gut dysbiosis-derived neuroinflammation in exacerbating the onset of neurodegenerative and neuropsychiatric disorders, opening up novel directions to explore gut–brain crosstalk and its underlying molecular mechanisms. It is noteworthy that systemic inflammation triggered by bacterial products in the blood stream may stimulate a strong production of pro-inflammatory cytokines by cells from the innate and the adaptive immune system, which can spread through the blood, promote the permeabilization of the BBB and reach the brain. Accordingly, CD4^+^ T-cells, when stimulated, release high levels of NF-κB, chemokines and inflammatory cytokines (e.g., TNF-α, IL-1β, IFN-γ and IL-6) and microglial activation markers (e.g., glial fibrillary acidic protein (GFAP), Sox-10 and TLRs), which in turn produce increased levels of glutamate, TNF-α and reactive oxygen and nitrogen species (ROS/RNS), inducing neuronal death and the further generation of oxidized and nitrated proteins, thereby indicating a tight association between neurodegenerative disorders and gut inflammation [70]. In this scenario, the nutritional therapeutic approach using hormetic nutrients, including polyphenols, in synergy with probiotics is attracting considerable interest in the scientific community to prevent and treat inflammation associated to pathophysiological changes in the diversity of the gut microbiota leading to nervous system disorders along the gut–brain axis [71,72]. Of particular interest, probiotics in combination with polyphenols induce a pharmacokinetic resilience to peripheral inflammation that ultimately protects against brain neuroinflammation by modifying immune cell recruitment to potentially preserve BBB integrity and prevent cognitive dysfunction under chronic stress conditions [73]. Importantly, microbial-derived metabolites modulate the intestinal immune response via the activation of aryl hydrocarbon receptor (AHR) to promote resilience to stress-induced psychiatric disorders by evoking adaptive changes [74]. Notably, it has been documented that a grape-derived prebiotic, in synergistic combination with the probiotics *Lactobacillus plantarum* and *Bifidobacterium longum*, acts as a ligand to the AHR on antigen-presenting cells or directly on the naïve CD4^+^ T cells, impacting their response to stress and ultimately leading to a reduction in the proportions of the Treg and Th17 lymphocytes in the periphery, in particular in the ileum and liver. The resilience response to inflammation in the periphery ultimately protects against neuroinflammation in the brain (i.e., pre-frontal cortex and hippocampus) by altering the recruitment of immune cells and potentially protecting BBB integrity in gut–brain axis disorders [74]. In addition, magnolol, a lignan found in *Magnolia officinalis* at a dose of 10 mg/kg, effectively increased the serum levels of tryptophan metabolites including kynurenic acid, 5-hydroxyindoleacetic acid, indole-3-acetic acid, indolelactic acid and indoxylsulfuric acid, which are endogenous AHR ligands to reduce pro-inflammatory cytokines (TNF-α, IL-6 and IL-1β) and attenuate dextran-induced colitis in rodents [75]. Recent preclinical research described curcumin as capable of suppressing the activation of the TLR4/NF-κB/STAT pathway that regulates the expression of pro-inflammatory genes [71]. In addition, a treatment with curcumin nanoparticles significantly decreased the mucosal mRNA expression of TNF-α, IL-1β, IL-6, CXCL1 and CXCL2 in colonic epithelial tissues [76]. Overall, according to hormesis, the dose is a crucial determinant in promoting gastro- and neuroprotective or harmful effects of nutritional therapies, in particular with polyphenols alone and/or in synergistic combination with probiotics; therefore, this must be carefully assessed. 

### 2.1. Resveratrol

The 3,5,4-trihydroxy-trans-stilbene also known as resveratrol is a natural polyphenolic compound produced by various plants in response to injury or pathogens such as bacteria or fungi. Food sources of resveratrol include the skin of grapes, berries and red wine [77]. It promotes numerous pharmacological properties including antioxidant, anti-inflammatory, anti-aging and anti-carcinogenic effects, gut microbiota homeostasis, neuroprotection, xenohormetic effects and resilience to stress. In congruence with this, pre-clinical evidence has shown that resveratrol is neuroprotective via modulating the intestinal flora and gut–brain axis dose-dependently [78]. Interestingly, gut bacteria and, in particular, *Bifidobacterium infantis* and *Lactobacillus acidophilus*, convert the glycosidic form, resveratrol-3-O-β-d glucoside, also called piceid, into the bioactive form, *trans*-resveratrol [79,80]. Resveratrol and its derivative pterostilbene have been shown to protect against intestinal morphological alterations, increased gut permeability, redox imbalance and dysbiosis. In particular, the molecular mechanism of action of resveratrol on the gut microbiota is its ability to act on duodenal-mucosal Sirt1 and, thus, improve insulin sensitivity and lower hepatic glucose levels [81]. In addition to its action on duodenal mucosa, resveratrol improves hypothalamic insulin sensitivity [81] and mitochondrial biogenesis [82], upregulating Sirt1 via the gut–brain axis. Furthermore, a study conducted by Chen et al. showed that a resveratrol-supplemented diet (300 mg/kg) modulates the microbiota by inhibiting the abundance of *Escherichia* and *Actinobacillus* and increasing the growth of *Bacteroides*, *Lactobacillus* and *Bifidobacterium* after 2 weeks in piglets [83]. Furthermore, resveratrol regulates the gut microbiota by activating anti-inflammatory lipid mediators including lipoxin A4, resolvins, protectins and maresins [84]. Intriguingly, resveratrol confers neuroprotection via modulating intestinal immune dysfunction. Specifically, the health effects induced by resveratrol in the brain are attributed to promoting the Th1/Th2 balance towards Th2 polarization and skewing the Treg/Th17 balance towards Treg in the SI-LP and subsequently reducing small intestinal pro-inflammatory cytokines (IL-17A, IL-23, IL-10, IFN-γ and IL-4) and intestinal vascular permeability via attenuating the circulating levels of inflammatory cytokines from the small intestine and alleviating cytokines-mediated BBB disruption and neuroinflammation in mice with a focal cerebral ischemia/reperfusion injury [85]. Of equal importance, resveratrol regulates the balance of neurotransmitters such as BDNF and serotonin 5-hydroxytryptamine (5-HT) via the GLP-1 pathway and the activation of *Sirt1* and *Foxo* genes in the intestine and CNS [86]. Indeed, several authors demonstrated that resveratrol exhibits anxiolytic and antidepressant effects similar to drugs in a range from 10 mg/kg to 80 mg/kg per day in a dose-dependent manner, upregulating both ERK and BDNF, as well as antioxidant markers (i.e., SOD and catalase), and downregulating inflammatory markers in animal models [87,88,89]. Unfortunately, few clinical trials have evaluated the health-promoting effects of resveratrol in gut–brain axis disorders. A recent study of ten participants with a mild decline in cognition demonstrated that 72 g of grape consumption for a 6-month period was related to a marked protection from longitudinal changes in brain metabolism and cognitive function, with an amelioration in attention and working memory performance [90]. In addition, another human study of 12 healthy volunteers observed interindividual differences in *trans*-resveratrol metabolism closely related to the microbial diversity in feces samples. In particular, the study showed that an oral dose of 0.5 mg of *trans*-resveratrol induced lower abundances of Firmicutes and higher abundances of *Bacteroidetes*, *Actinobacteria (Bifidobacteriaceae* and *Coriobacteriaceae)*, *Verrucomicrobia* and *Cyanobacteria*. The authors identified two new resveratrol metabolites, 3,4′-dihydroxy-trans-stilbene and 3,4′-dihydroxybibenzyl (lunularin), produced by intestinal bacteria [91]. Furthermore, a recent double-blind, placebo-controlled randomized clinical study on sixty-two participants demonstrated that resveratrol at a dosage of 250 mg twice per day in synergy with risperidone reduced hyperactivity and secondary deficits such as social withdrawal, stereotypic behavior and inappropriate speech in patients with ASD compared to a placebo group [92]. Lastly, a randomized, placebo-controlled, double-blind, multicenter 52-week phase 2 study demonstrated that resveratrol at an oral dose of 500 mg once daily was able to permeate the BBB, acting on the CNS and increasing brain volume loss in patients with mild to moderate AD [93].

### 2.2. Curcumin

Curcumin is a polyphenolic compound present in *Curcuma longa* Linn rhizome (Zingiberaceae). The recent interest in its therapeutic potential derives from numerous biological effects including antioxidant [94], anti-inflammatory [95], gut microbiota homeostasis [96] and neuroprotection [97] in a dose-dependent manner. Emerging preclinical and clinical literature reported the health-promoting effects of curcumin in gut and brain disorders [98,99]. In the gut microbiota, a treatment of curcumin significantly altered the ratio between beneficial/pathogenic bacterial strains by increasing the growth of *Bifidobacteria* and *Lactobacilli* and reducing the loads of *Prevotellaceae*, *Coriobacterales*, *Enterobacteria* and *Enterococci* [100]. Indeed, a dose of 500 mg of curcumin and boswellia extracts significantly improved bloating and abdominal pain symptoms in patients suffering from irritable bowel syndrome (IBS) and small bowel dysbiosis after 30 days of supplementation [101]. In the CNS, curcumin provided neuroprotection, ameliorating motor deficits, the aggregation of α-synuclein and microglial activation with a remarkable abundance of *Lactobacillacea* and *Lachnospiraceae*, increasing the serum levels of methionine, tyrosine, sarcosine and creatine, and depleting the growth of *Aerococcaceae* and *Staphylococcaceae* in a mouse model of PD [97]. In particular, the neuroprotection induced by curcumin resulted in the upregulation of redox sensory genes such as HO-1 and NQO1 via the Nrf2-ARE pathway [102]. Moreover, low doses (20 and 40 mg/kg) of curcumin decreased the sensitivity of the intestinal tract-produced antidepressant- and anti-anxiety-like effects by elevating serotonin, BDNF and p-CREB levels in the hippocampus in a rat model of IBS [103]. In humans, a randomized controlled study conducted by Wynn et al. demonstrated the effectiveness of curcumin capsules at a dose of 360 mg/day (twice daily as an oral dose) in ameliorating BDNF levels and enhancing cognitive performance in patients with schizophrenia after 8 weeks [104]. In addition, a randomized, double-blind trial of 28 participants proved that oral supplementation of curcumin (2.5 g) for three months was able to decrease uremic toxins such as p-cresyl sulfate plasma levels in hemodialysis patients, modulating the gut microbiota [105]. Collectively, curcumin could represent a potential therapeutic option against brain disorders through the regulation of intestinal dysbiosis, thus improving the abundance of beneficial bacteria and maintaining a proper gut–brain axis without any apparent toxicity.

### 2.3. Blueberry

Blueberries are polyphenolic compounds rich in anthocyanins, fiber and sugars with powerful antioxidant and anti-inflammatory properties capable of modulating the gut microbiota and brain function in a biphasic dose response manner. It is noteworthy that blueberry positively influences gut microbiota composition, exhibiting antimicrobial and antiadhesion properties against pathogenic bacteria and ultimately contributing to brain health [106,107]. Accordingly, recent preclinical evidence reported that the anthocyanin-rich extract obtained from blueberries at a dose of 30 mg/day (i) reduced neuroinflammation and gut dysbiosis by modulating gut microbiota composition, especially increasing the abundance of *Lactobacillales* and decreasing the *Clostridiales* population; (ii) enhanced serotonin levels in the cerebral prefrontal cortex and intestine; and (iii) reversed the synaptic dysfunction in rodent models of autism-like behaviors [106]. In addition, a dose of 300 mg/kg per day of polyphenol-rich blueberry–mulberry extract significantly changed the gut microbiota by enhancing the abundance of *Lactobacillus*, *Streptococcus* and *Lactococcus*, and decreasing the abundance of *Blautia* and *Allobaculum*. This was positively correlated with an increase in beneficial metabolites, such as α-linolenic acid, vanillic acid and N-acetylserotonin, to alleviate inflammation and cognitive impairment along the gut–brain axis in aged mice after 6 weeks of treatment [107]. Furthermore, low doses (6%) of blueberry anthocyanin-rich extracts attenuate high-fat diet-induced oxidative stress and improve hippocampal status, as well as neuronal function, through an increase in antioxidant enzymes (i.e., SOD and GSH-Px). Notably, the modulation of the antioxidant pathway stimulated probiotics (*Bifidobacterium* and *Lactobacillus*) and SCFAs producers (*Roseburia*, *Faecalibaculum* and *Parabacteroides*), improving the colon environment in C57BL/6 mice [108,109]. Human studies showed that the consumption of a wild blueberry powder beverage in men increased live *Bifidobacterium* activity after six weeks [110]. Moreover, a pilot study conducted on 17 women reported that a low dose of 38 g of polyphenol-rich fractions purified from blueberry (i.e., anthocyanins/flavonol glycosides, sugar/acid fraction, proanthocyanidins and total polyphenols) and/or prebiotic mix modulated the fecal microbiota composition of healthy adults and was correlated with an increase in the antioxidant activity in blood. The study observed that the fecal microbiota fermented with total polyphenols and sugar/acid fraction supplementation exhibited a higher abundance of *Enterobacteriaceae* and *Bifidobacteriaceae*, while anthocyanins/flavonol glycosides supplementation and prebiotic mix produced a significant reduction in *Escherichia*/*Shigella (Enterobacteriaceae)* and an increase in the *Lachnospiraceae* and *Bacteroidaceae* families [111]. In addition, a daily supplementation with blueberry powder (0.5 c) whole-fruit equivalent improved cognitive performance in middle-aged patients with insulin resistance after 12 weeks [112]. Similarly, supplementation with blueberry powder at a dosage of 12.5 g per packet enhanced the brain response and memory function in older adults with MCI and a risk of dementia [113]. Furthermore, a double-blind, parallel randomized controlled trial conducted in 61 healthy older adults supplemented with 26 g of freeze-dried wild blueberry showed significant improvements in vascular and cognitive function, as well as in the gut microbiota composition, by increasing beneficial bacteria such as *Ruminiclostridium* and *Christensellenacea* [114]. Another randomized, placebo-controlled clinical study reported that blueberry intake (25 g/day) during a period of 18 days markedly decreased the plasma concentrations of pro-inflammatory 9,10-, 12,13-dihydroxy-9Z-octadecenoic acids (diHOMEs), increased anti-inflammatory docosahexaenoic acid (DHA)- and eicosapentaenoic acid (EPA)-generated hydroxydocosahexaenoic acids and specialized pro-resolving oxylipins after intense exercise in untrained adults compared to a placebo [115]. Finally, a recent double-blind, randomized, cross-over study reported that 30 g of highbush freeze-dried blueberry powder produced from equal proportions of Tifblue^®^ and Rubel^®^ varieties (equivalent to 180 g fresh blueberries) significantly attenuated abdominal symptoms and ameliorated general markers of well-being, quality of life and life functioning compared to placebo in patients with gastrointestinal disorders after six weeks of treatment [116].

### 2.4. Hidrox^®^

Hidrox^®^ (HD) is a freeze–dried powder extract from the aqueous fraction of olives obtained from defatted olive pulps, during the processing of *Olea europaea* L. after olive oil extraction [117]. An amount of 12% HD extract contains polyphenols with high antioxidant potential. The most abundant polyphenol in HD is hydroxytyrosol (40–50%), while 5–10% contains oleuropein, approximately 20% oleuropein aglycone and gallic acid and 0.3% tyrosol [118]. Recent preclinical evidence demonstrated that HD possesses significant antioxidant and anti-neuroinflammatory effects by upregulating the Nrf2 pathway and *vitagenes* and downregulating NF-κB signaling to prevent or delay the neurodegenerative process characteristic of AD and PD [119,120]. Furthermore, other recent evidence showed that low doses (10 mg/kg) of HD modulate oxidative stress and neuroinflammation through a significant reduction in proinflammatory mediators, such as IL-1β, IL-6 and TNF-α, and a significant induction of the Nrf2/HO-1 pathway by restoring GSH, SOD and catalase levels in the bladder and spinal cord of rodent models [121]. In addition, studies on *Caenorhabditis elegans* used as model of PD have reported that low doses of 250 μg of HD display healthy effects by improving lifespan and stress resistance, mainly exerting a neuroprotective action by reducing neurotoxic misfolded α-synuclein aggregates in dopaminergic neurons [122,123]. Clinical research documented the health effects of hydroxytyrosol on cognitive function. Notably, a randomized, double-blind, placebo-controlled, parallel-group study on 72 subjects showed that 3 g twice daily of desert olive tree pearls (DOTPs) containing 162 times more polyphenol hydroxytyrosol than olive oil, improved cognitive function (memory, attention, reaction time and executive function) in middle-aged and older adults [124]. Furthermore, in the intestine, hydroxytyrosol supplementation exerts anti-inflammatory effects in dextran sodium sulfate-induced ulcerative colitis by augmenting the colonic Nrf2 pathway, inhibiting NLRP3 inflammasome activation (interleukin-18 and interleukin-1β levels) and modulating gut microbiota in rodents [125,126]. It is interesting to note that an oral administration of 100 mg/kg body weight HT modulated colonic gut microbiota by increasing the number of *Firmicutes* and *Lactobacillus* and reducing the relative amount of *Bacteroidetes.* The gut beneficial effects were correlated with the activities of serum Nrf2 antioxidant enzymes (i.e., SOD, GSH-Px, catalase and NQO1) in the small intestine of mice [127]. Finally, another randomized, controlled, double-blind, crossover human study of 12 participants showed that the ingestion of virgin olive oil (25 mL/day) containing a mixture of olive oil and thyme phenolic compounds (500 mg) determined changes in the gut microbiota by increasing *Bifidobacteria* populations, decreasing blood-LDL in hypercholesterolemic patients after 3 weeks [128]. Overall, hidrox^®^ can be taken into consideration as a novel approach of therapeutic intervention for intestinal and brain disorders as it leads to changes in gut beneficial bacteria and ultimately enhances the cognitive function and quality of life in humans. 

### 2.5. Crocus sativus L.

*Crocus sativus* L., widely known as saffron, is a perennial plant belonging to the Iridaceae family. Commonly, it is cultivated in Iran, Spain, Morocco, Turkey, India, Greece and Italy [129]. The flowers comprise six purple tepals where both petals and sepals are fused; moreover, they present a white style surrounded by three yellow stamens and a red stigma separated into three threads. Saffron performs several pharmacological activities such as antioxidant [130], anti-inflammatory [131], hepatoprotective [132], neuroprotective [133], antidepressant [134] and anti-carcinogenic activity [135], and maintains gut microbiota balance [136]. Recent literature documented the heathy effects of saffron in gut–brain axis disorders. Notably, it has been documented that there are significant antibacterial dose-response effects of saffron polyphenols in the reduction of *Lactobacillus* and *Clostridium* species in the cecal microbiome in vivo [136]. The major bioactive molecules of saffron include crocin (C_44_H_64_O_24_), crocetin (C_20_H_24_O_4_), picrocrocin (C_16_H_26_O_7_) and safranal (C_10_H_14_O). Interestingly, a study conducted by Khodir et al. showed the anti-inflammatory and anti-oxidant activity of crocin at a low dose of 20 mg/kg orally via the upregulation of Nrf2/HO-1 signaling and the downregulation of caspase-3 activity in the colon of rats [137]. Furthermore, safranal inhibited gut tissue damage induced by dextran sodium sulfate, ROS and intestinal epithelial cell death, showing significant protective effects in maintaining intestinal homeostasis in drosophila [138]. Furthermore, a recent study demonstrated that a new herbal formulation containing *Edgeworthia gardneri* (Wall.) Meisn., *Sibiraea angustata* and *Crocus sativus* L., at a total dose of 26 g, improved hyperglycaemia and modulated gut dysbiosis by reducing lipopolysaccharide (LPS) levels; the same study also reported a dose-dependent decrease in circulating levels of IL-6 and TNF-α along with an increase in the amount of *Proteobacteria* and *Actinobacteria* and a reduction in the *Firmicutes*/*Bacteroidetes* ratio in diabetic fatty rats after 6 weeks [139]. In the brain, saffron constituents have been shown to be neuroprotective by upregulating Nrf2 signaling and stress resilience *vitagenes* in numerous brain disorders, including neurodegenerative and neuropsychiatric pathologies, in a hormetic dose response manner [133]. In line with this, saffron (40 mg/kg) was synergistically associated with endurance exercise, increased BDNF, serotonin and muscular neurotrophin-3 mRNA levels in the hippocampus of rats [140]. Furthermore, the neuroprotective efficacy of saffron tea infusion (90 mg styles/200 mL) against aflatoxin B1-induced neurotoxic damage in terms of enhancing learning and memory performance and cholinergic activity, as well as reducing monoaminergic and oxidative markers in the brain of rodent models after 2 weeks, has also been demonstrated [141]. Further evidence revealed that a low dose of 30 mg/kg via oral gavage of crocin improved unpredictable chronic mild stress-induced anxiety and depression through a downregulation of brain oxidative stress, cortical malondialdehyde, inflammatory mediators (i.e., TNF-α and IL-6) and corticosterone serum levels after 4 weeks in rats [142]. Finally, a clinical trial conducted by Kell and coworkers on 128 participants with low mood showed that affron^®^ at a dosage of 28 mg per day for 4 weeks improved mood, anxiety and stress management without side effects compared to a placebo group [143]. Overall, the data suggest that saffron and its active compounds show an excellent safety profile and could represent potential candidates for the prevention and treatment of gut and brain disorders.

### 2.6. Polyphenols of Nutritional Mushrooms

The recent literature has shown great attention to the beneficial effects of nutritional mushroom polyphenols contained in *Hericium erinaceus* and *Coriolus versicolor* for treating gut microbiota dysbiosis and cognitive damage [144,145]. Notably, extensive evidence documented that the polyphenols of HE and CV inhibit neuroinflammation and oxidative stress to prevent and/or slow the onset of major neurodegenerative diseases, including AD and PD [55,146,147], and psychiatric disorders such as ASD [148].

#### 2.6.1. *Hericium erinaceus*

*Hericium erinaceus* (HE) is a medicinal mushroom containing several bioactive constituents with anti-cancer, anti-inflammatory, anti-oxidative and neuroprotective properties [55,148,149]. In particular, HE contains erinacines and hericerin that induce a neurotrophic effect by stimulating the nerve growth factor (NGF) biosynthesis responsible for the survival, growth and differentiation of neurons [150]. Interestingly, HE also contains a great number of polyphenols acting as strong antioxidants for the inhibition of tyrosinase and free radical scavenging [151]. Preclinical evidence showed that HE can be used as a prebiotic to regulate the gut microbial community in animal models [152]. In particular, it has been demonstrated that an intake with 0.8 g of HE for 16 weeks can regulate the gut microbial community including the pathogenic intestinal genera *Campylobacter*, *Streptococcus* and *Tyzzerella* implicated in inflammation and obesity in aged dogs [152]. Moreover, the supplementation of HE improved gut microbiota composition, producing SCFAs both in rats with ulcerative colitis [153] and in healthy adults [154] to enhance the gut barrier function. In the brain, HE attenuated p-tau and amyloid-β deposition while in the intestine it improved gut microbiota diversity by promoting the growth of SCFAs-producing bacteria and by suppressing the abundance of *Helicobacter*. In the same study, HE was also shown to enhance the activation of superoxide dismutase, catalase and glutathione peroxidase, and inhibit the secretion of malondialdehyde and 4-hydroxynonenal, confirming that in APP/PS1 mice this mushroom promotes neuroprotective effects via upregulation of the Nrf2 pathway [155]. Furthermore, HE influenced the composition of the intestinal microbiota, particularly increasing the relative abundance of beneficial bacteria such as *Lachnospiraceae*, *Ruminococcaceae* and *Akkermansiaceae*, and reducing the microbial population of *Muribaculaceae*, *Rikenellaceae*, *Lactobacillaceae* and *Bacteroidaceae*, promoting the immunomodulatory effects via the NF-κB, MAPK and PI3K/Akt pathways in vitro and in experimental animal models after 28 days [156]. In addition, HE *mycelium*-derived polysaccharide supplementation at a dose of 500 mg/kg per day markedly improved the nutritional status and decreased the incidence of diarrhea and intestinal inflammation through an increase in *Lactobacillus reuteri* and a reduction in *Streptococcus lutetiensis* in cynomolgus monkey used as a model of ulcerative colitis after 4 weeks [157]. In addition, other authors also demonstrated that a new polysaccharide isolated from HE (HEP10) suppressed the LPS/DSS-induced production of iNOS and COX-2 and proinflammatory mediators such as TNF-α, IL-6 and IL-1β, as well as NLRP3 inflammasome, in rodent models of colitis and in macrophages. Furthermore, HEP10 reversed the dextran sulfate sodium (DSS)-induced changes in both composition and structure of intestinal bacteria by reducing the Proteobacteria phylum and increasing *Akkermansia muciniphila* dose-dependently [158]. Finally, a recent pilot trial conducted by Xie et al. on 13 healthy individuals observed that 3 g/day of HE powder taken as a food supplement modulated gut microbiota composition and serum biochemical parameters. Notably, the study found that HE significantly upregulated beneficial bacteria, such as *Bifidobacterium* and *Bacteroidetes*, as well as SCFAs-producing bacteria, such as *Roseburia* and *Faecalibacterium*, and downregulated pathobionts, such as *Escherichia_Shigella*, *Streptococcus thermophilus*, *Bacteroides caccae* and *Anaerostipes hadrus*, in blood and fecal samples [159].

#### 2.6.2. *Coriolus versicolor*

*Coriolus versicolor* (CV) is another nutritional mushroom that acts as a prebiotic to modulate the intestinal microbiome composition [160] and brain function [60]. Recent evidence reported the anti-inflammatory and antioxidant properties of CV in experimental models [161,162,163]. In line with this, it has been shown that CV at a dose of 200 mg/kg induced antioxidant and anti-inflammatory effects, upregulating the Nrf2/HO1 pathway and downregulating the Toll-like receptors 4 (TLR4)/NF-kB cascade in a murine model of colitis [161]. Interestingly, the synergistic combination of HE and CV at a dose of 200 mg/kg orally for 28 days upregulated antioxidant Nrf2/HO-1 redox signaling and increased anti-inflammatory lipoxin A4 to inhibit the neuroinflammation (NF-κB pathway) and apoptosis of dopaminergic neurons in a rodent model of PD [162] and TBI [147]. In addition, a very recent clinical study by our research group demonstrated that a CV treatment for 6 months at a moderate dosage promoted neuroprotective effects by downregulating α-synuclein and NF-κB-mediated pro-inflammatory cytokines and upregulating the Nrf2 pathway and stress resilience of *vitagenes* to counteract oxidative damage, particularly of protein carbonyls and 4-hydroxynonenal (4-HNE), in the lymphocytes of patients with Meniere’s disease and at increased risk of developing neurodegenerative disorders [163]. Finally, more recent evidence has shown that CV improves gut dysbiosis, predominantly suppressing *Clostridium* (belonging to *Firmicutes*) and increasing the *Bacteroides* phylum both in the serum and cecal contents [164]. 

## 3. Allostasis and Resilience in Gut and Brain

Stress adaption to environmental challenges is also defined as “allostasis”. The concept of allostasis emerges as a new view of stress and resilience to it, specifically referring to the process that maintains cellular homeostasis and drives other physiological changes in response to environmental perturbations [165]. Allostasis responses are not considered negative but reflect the fact that stress adaptive responses are crucial for survival, healthspan and lifespan extension [165]. The gut and brain are central organs of oxidant stress and resilience/adaptation to stress (allostasis), but they also contribute to pathophysiology (‘‘allostatic load/overload’’) when they are overused or perturbated. Brain resilience denotes the capacity of an organism to cope to neuronal insults (e.g., neurotoxicity, neuroinflammation and oxidative stress) or psychological perturbations through the activation of allostasis mechanisms and neuronal mediators to promote stress adaptation and mental health. In line with this, the neuronal mediators of stress adaptation implicated in gut and brain homeostasis—such as HPA; cortisol; the autonomic nervous system and vagus nerve; metabolic hormones and immune system mediators; neurotrophins, such as BDNF; endogenous neurotransmitters such as GABA, glutamate, acetylcholine and serotonin; cellular stress response mediators such as Nrf2 and *vitagenes*; and anti-inflammatory lipid mediators (lipoxin A4)—promote adaptation to several stressors (e.g., oxidative stress, inflammation and mitochondrial disfunction), but the same mediators display biphasic effects and can also participate in pathophysiology when they are dysregulated or overused with respect to their physiological balanced network (allostasis load and overload) [166]. Therefore, acute stress, also defined as “eustress”, is beneficial in the short term since it enhances the adaptive capabilities of an organism and increases the antioxidant resilience responses, hormesis and cross-tolerance mechanisms, and is considered an example of allostasis; however, in the long term it turns into chronic stress, also called “distress”, and is maladaptive, leading to a dysregulated Nrf2 pathway and predisposing one to the onset and progression of central nervous system disorders [167]. For instance, proinflammatory cytokines can modulate the production of corticosteroids, which in turn can suppress inflammatory cytokine production. Likewise, the sympathetic and parasympathetic systems exert different effects on inflammatory cytokines, with the former stimulating their production and the latter inhibiting them. It is important to emphasize that allostasis and allostatic load and overload apply to the gut–brain axis [168]. An optimal gut equilibrium due to the amounts of beneficial bacteria and their metabolites promotes functional neural activity that drives synaptic plasticity, mediated in part by the vagus nerve and systemic hormones, but also by endogenous excitatory and inhibitory neurotransmitters, amino acids, neurotrophic factors and other mediators. Changes in how such adaptive mediators respond likely explain the changes in the vulnerability of the gut microbiome to environmental stressors such as proinflammatory cytokines produced by pathogenic bacteria, leading to intestinal dysbiosis or favoring mental illnesses including AD, PD, depression and autism. Indeed, the gut and brain are constantly adapting to a changing environment. This is the very essence of allostatic load/overload applied to the gut and brain [168]. Therefore, allostatic overload results in dysregulated HPA axis regulation, elevated chemokines and inflammatory cytokines, reduced synaptic plasticity and persistently activated enteric microglia, culminating in a dysbiotic gut microbiota. The functional crosstalk between the gut–vagal sensory neurons–brain axis is also implicated in allostasis for stress adaptation and resilience [168]. Several independent pathways and bioactive molecules contribute to the gut–brain axis bidirectional signaling. These pathways include proinflammatory mediators, metabolic signaling, oxidative markers, stress modulators, dietary nutrients, neuroendocrine factors and a direct neuronal crosstalk via the vagus nerve [169,170]. Inflammatory processes protecting the body from injury or invasion also impose an allostatic load and the vagus nerve mediates the inflammatory reflex. Notably, afferent vagus sensory fibers sense peripheral inflammatory cytokines produced by gut bacteria and convey signals to the brain to generate an adaptive or maladaptive response. The latter triggers a neuroinflammatory cascade and dysregulated BBB, culminating in impaired cognitive function. On the other hand, inflammatory signals activate the hypothalamus via vagal afferents, triggering cholinergic vagal efferent fibers to suppress the release of local and serum proinflammatory cytokines and macrophages, restoring the control mechanisms of the adaptive anti-inflammatory reflex [171]. Importantly, afferents and efferent vagal sensory fibers in allostasis are implicated in modulating the potentially detrimental effects of allostatic load/overload for gut and brain health. Importantly, allostatic load is a cumulative measure of physiological dysregulation and is influenced by multiple factors including nutrition. Emerging evidence reported that berries intake and their bioactive polyphenols alleviate stress by modulating the BDNF and HPA axis in the hippocampus, which in turn reduces neuroinflammation and attenuates allostasis load scores, ultimately improving resilience and potentially reducing the severity of stress-related disorders [172]. Similarly, probiotics are known to affect the composition and function of microbiota for the maintenance of allostasis adaptations, ultimately promoting brain resilience in the context of environmental stress challenges related to perturbations of gut microbial diversity (dysbiosis), leading to systemic inflammation/neuroinflammation [170]. In the same way, the ingestion of a strain of *Lactobacillus* in mice has been shown to attenuate stress-induced neurogenic skin inflammation (allostasis) and the inhibition of hair growth along the skin–gut–brain axis [173]. Overall, the antioxidant nutritional approach in synergy with psychobiotics targeting crucial pathways and neuronal mediators including the Nrf2 pathway to modify the composition of the gut microbiota and stimulate vagal sensory neurons could provide an effective alternative treatment paradigm to promote allostasis and resilience during stressful challenges in gut–brain axis disorders.

## 4. The Role of Epigenomics and Nutrition in Gut–Brain Axis

Epigenomics has grown rapidly over the past two decades due to its importance in numerous physiological and pathophysiological processes, including gut–brain axis disorders [174]. Epigenetics consists of DNA methylations, histone modifications and the dysregulation of micro-RNAs in order to regulate the accessibility of specific DNA regions to transcription factors and allow them to adapt the genomic expression to the environmental changes [175]. Emerging evidence demonstrated that epigenetics mediates gene–environment interactions and contributes to the vulnerability of neurodegenerative and psychiatric disorders such as depression and anxiety, schizophrenia and autism [176]. Accordingly, studies documented epigenetic aberrations such as DNA methylations, histone modifications or miRNAs dysregulations found in the postmortem brain or blood cells of patients with psychotic or autism-spectrum disorders tightly linked to the pathogenesis [177]. Epigenetics plays a crucial role in regulating host physiology by altering the metabolic activity of the intestinal microbiome, which depends on the environment and nutrition. Interestingly, gut microbiota bacteria synthesize and modulate their hosts to produce metabolites, such as SCFAs [178], tryptophan and indole derivatives [179] and neurotransmitters including GABA, glutamate, acetylcholine, dopamine, norepinephrine [180], Nrf2 and polyphenols [181]. The epigenetic interactions between the gut–brain axis via several bacterial-produced metabolites and cytokines and their ability to cross the BBB allow them to explicate their neuroactive properties that are potentially implicated in mental health and/or disease. Importantly, SCFAs are important mediators by which bacteria affect the host epigenome and can inhibit the activity of histone deacetylases (HDACs), resulting in chromatin changes generally associated with increased histone acetylation of non-coding target genes [182]. Among SCFAs, butyrate and propionate produced by commensal microbials induced the differentiation of T_reg_ cells and ameliorated intestinal inflammation and the development of colitis via an enhancement of histone H3 acetylation in the promoter of naïve T cells and expression of the Foxp3 gene in vitro and in vivo [183]. Of relevance, changes in DNA methylation have been ascribed to an increase in pro-inflammatory cells, in particular promoters of IL-6 and IL-1β, as well as promotors of chemokine and chemokine receptors, such as CXCL13 and CXCR3, with significantly hypomethylated and elevated levels of neutrophils found in stools of children affected by ASD specifically associated with dysbiosis, an inflammatory state and gut permeability features [184]. Nowadays, nutritional therapy through polyphenols and/or probiotics that interact with DNA, also referred to as “nutriepigenomics”, represents a complementary and alternative approach in gut and nervous system disorders, demonstrating positive effects in subjects with epigenetic variations and drug resistance [185].

### 4.1. Tryptophan, Kynurenine and Indole Pathways

Tryptophan metabolism along the serotonin, kynurenine and indole pathways can be directly influenced by gut microbials. Some bacteria strains (i.e., *Escherichia coli*, *Clostridium* sp. and *Bacteroides* sp.) harbor a tryptophanase enzyme responsible for the conversion of tryptophan into indole and its derivatives, such as indole-3-aldehyde (IAld), indole-3-acetic-acid (IAA) and indole-3-propionic acid, which can give rise to a wide range of neuroactive signaling molecules. Additionally, tryptophan can be metabolized into 5-HT via aromatic amino acid decarboxylase (AAAD) activity, or kynurenine by the enzymes tryptophan-2,3-dioxygenase (TDO) or the ubiquitous indoleamine-2,3-dioxygenase (IDO). Lipopolysaccharides (LPS), an inflammatory cell wall component from Gram-negative bacteria, can induce the expression of IDO, increasing the conversion of tryptophan to kynurenine. The latter, produced from the periphery, is also a potent agonist of AHR capable of crossing the BBB to regulate the intestinal immune system [186]. On the other hand, circulating SCFAs, such as butyrate, can directly modulate central kynurenine pathways. Specifically, butyrate has been demonstrated to inhibit IDO activity and to modulate the kynurenine pathway in a STAT1/HDAC-dependent manner [187]. In addition, in stressed mice, the production of H_2_O_2_ by Lactobacillus strains, such as *Lactobacillus reuteri*, can be protective against the development of despair behaviors by directly inhibiting intestinal indolamine 2,3-dioxygenase 1 (IDO1) expression and decreasing the circulating kynurenine levels [187]. Conversely, the reduction in Firmicutes and the subsequent decrease in SCFAs synthesis observed in MDD patients has been linked to increased inflammation, and cytokines are also known to promote tryptophan utilization for kynurenine synthesis via IDO activity [188]. This pathway gives rise to the neurotoxic metabolite quinolinic acid that crosses the BBB to reach the CNS and reduces central serotonergic availability [189]. Clinical research showed that children with ASD exhibit an altered tryptophan metabolism, with reduced levels of tryptophan and an increased kynurenine to tryptophan ratio in the plasma [190], suggesting a shift in the tryptophan metabolism from serotonin synthesis to the kynurenine pathway. Furthermore, in these ASD children, a correlation was observed between altered concentrations of tryptophan and serotonin with gut dysbiosis thorough a significant decrease in *Dorea* and *Blautia*, as well as *Sutterella*, and an increase in *Clostridiales* associated with the severity of the symptoms [190]. Probiotic species belonging to *Lactobacillus* and *Bifidobacterium* may shift the host tryptophan metabolism by suppressing the kynurenine pathway. Furthermore, *Lactobacillus* are reported to be able to degrade tryptophan into indole compounds, such as IAld, ILA and IAA [191]. In addition, the administration of *Lactobacillus rhamnosus* leads to the remodeling of the DNA methylation code at the *BDNF* and *Tph1A* promoter genes in the gut and brain of zebrafish after 28 days [192]. Recent evidence reported that indole-3-lactic acid can regulate gut homeostasis. Notably, an Escherichia coli strain isolated from the feces of tinidazole-treated individuals showed that indole-3-lactic acid reduced the susceptibility of mice to dextran sulfate sodium-induced colitis by inhibiting the production of epithelial CCL2/7, thereby reducing the accumulation of inflammatory macrophages in vitro and in vivo [193]. Similarly, the supplementation of polyphenols extracted from Fu brick tea (post-fermented tea) promoted the transformation by gut microbiota bacteria of tryptophan into indole-3-acetic acid attenuating colitis, immune cells infiltration and inflammatory cytokines release through a direct enhancement of AHR-mediated protection in rodents, dose-dependently [194]. The dietary enrichment of fibers from oat and rye brans enhanced the production of SCFAs, leading to improved gut integrity, reduced liver inflammation and the expression of genes related to tryptophan metabolism, in particular, tryptophan hydroxylase 1 (TPH-1) mRNA activity and enhanced indole propionic acid production in mice [195] and in humans [196]. 

### 4.2. γ-Aminobutyric Acid

γ-aminobutyric acid (GABA) is the main inhibitory neurotransmitter of the CNS. It is produced by various bacteria, including *Bacteroides*, *Bifidobacterium*, *Lactobacillus* and *Escherichia* spp., and this involves ENS homeostasis and disturbance, such as acid secretion, bowel motion, gastric emptying and abdominal pain [197]. Alterations in GABAergic gene promotors, such as GAD2, NPY and SST, involves H3K4 methylation as well as histone modifications leading to prefrontal dysfunction and the development of schizophrenia and autism [198]. Probiotic therapy with *L. rhamnosus* (JB-1) reduced GABA(Aα2) mRNA expression in the prefrontal cortex and amygdala, but increased GABA(Aα2) in the hippocampus, inhibiting stress-induced corticosterone and anxiety- and depression-related behavior in a rodent model via the vagus nerve [199]. Translational studies showed that *L. reuteri* PBS072 and *B. breve* BB077 are potential probiotic candidates for improving stress resilience, cognitive functions and sleep quality through the inhibition of the epigenetic enzyme LSD1, promotion of GABA and expression of serotonin [200]. Furthermore, probiotic-fermented buckwheat significantly increased the contents of GABA, rutin, total polyphenols and total flavonoids by improving oxidative stress and chronic inflammation, reversing the high-fat diet-induced intestinal dysbiosis through a reduction in the ratio of Firmicutes and Bacteroidetes and improving the abundance of SCFA-producing bacteria, such as *Bacteroides*, *Lactobacillus* and *Blautia*, in murine models of dyslipidemia [201]. Finally, a prenatal treatment with resveratrol-activating histone deacetylase Sirt1 prevented epigenetic alterations caused by valproic acid in the GABA receptor and synaptic proteins and the expansion of initial damage resulting in the maintenance of the neuronal composition in the medial prefrontal cortex and hippocampus in a rat model of autism [202].

### 4.3. Hypothalamic–Pituitary–Adrenal Axis

Epigenetic modifications are involved in neurophysiological changes, the inflammatory response, hypothalamic–pituitary–adrenal (HPA) axis activity and glucocorticoid receptor resistance, and they increase the risk for the development of metabolic and mental disorders [203]. Preclinical genetic evidence demonstrated that prenatal stress induces manic behavior in female offspring rats accompanied by hyperactivity of the HPA axis, epigenetic adaptations in the activity of HDAC and DNMT, and acetylation in the histones H3K9 and H3K14, as well as increased levels of adrenocorticotropic hormone (ACTH) [204]. Glucocorticoid receptor (NR3C1) is an important target for epigenetic regulation in the HPA axis. Importantly, decreased expression of the glucocorticoid receptor (*NR3C1*) gene, which is also susceptible to epigenetic modulation, is a strong indicator of impaired HPA axis control and function. Accordingly, recent evidence demonstrated that epigenetic changes driving the reprogramming of NR3C1 expression and regulation contribute to the aberrant expression profile of genes, particularly variations in *FKBP5* underlying the HPA axis impairment which likely decreases resilience to the adaptive stress response and increases susceptibility to psychiatric disorders and suicide in teenagers [205]. In addition, epigenetic variations of the *NR3C1* gene have also been associated with early-life experiences in both animals and humans. Studies of early experiences in rats showed that DNA methylation at CpG islands on the *NR3C1* promoter in specific regions of the brain was altered by maternal care, resulting in *NR3C1* expression and HPA responses to stress [206]. In humans, stressful life events (e.g., trauma and abuse) have been correlated with higher DNA methylation at the *NR3C1* promoter region and biological markers of HPA axis activity, such as salivary cortisol [207]. The gut microbial balance and/or a modified diet through probiotics supplementation and polyphenols may prevent or restore epigenetic alterations (e.g., proinflammatory genes in autism) [208]. Notably, children with autism and LPS-exposed rat model of autism exhibited lower SCFA concentrations and an overactivation of the HPA axis. Notably, SCFA-producing bacteria, such as *Lactobacillus*, might be the key differential microbiota between the control and LPS-exposed offspring [208]. Interestingly, a sodium butyrate treatment contributed to regulate the HPA axis, in particular corticosterone and corticotropin-releasing hormone receptor 2 expressions, ultimately improving anxiety and social deficit behaviors in autism-like rats [209]. Finally, a recent study demonstrated that a polyphenols-enriched diet, such as with quercetin (20 mg/day), xanthohumol (10 mg/day) and phlorotannins (20 mg/day), ameliorated the dysregulation of the HPA axis and monoamine neurotransmitters (i.e., dopamine and 5-hydroxyindoloacetic acid) and this was correlated with marked changes in bacterial composition and diversity, with a significant increase in *Enterorhabdus*, *Asteroplasma*, *Lachnospiraceae* and *Coprococcus*, ultimately reversing depression in animal models of early-life stress caused by maternal deprivation [210]. 

## 5. Polyphenol Nanoparticle Delivery Systems in Gut and Brain Disorders

Recently, several scientists have highlighted the importance of novel polyphenol nanoparticle delivery systems to enhance the bioavailability and stability of circulating polyphenols and ultimately preserve gut and brain health [211,212].

### 5.1. Polyphenol Nanoparticles for Brain Health

Nowadays, nanotechnology increasingly represents a promising approach for the delivery of therapeutic agents into the CNS, in particular polyphenol nanoparticles capable of improving the overall bioavailability and therefore diffusing more efficiently across the BBB to enhance brain delivery [213]. Preclinical evidence demonstrated that an intravenous injection of curcumin loaded with T807-modified nanoparticles at a dosage of 5 mg/kg was capable of effectively crossing the BBB by enhancing its permeation into the brain. Notably, curcumin loaded with T807-modified nanoparticles displayed a high binding affinity to the hyperphosphorylated form of tau protein in neurons by reducing its levels and inhibiting neuronal cell death to attenuate AD progression both in vitro and in vivo [214]. Moreover, other authors observed the neuroprotective potential of curcumin-loaded lipid-core nanocapsules in a murine model of AD. The results of this study showed that a low dosage of 10 or 1 mg/kg p.o. of curcumin-loaded lipid-core nanocapsules for 14 days provided higher neuroprotection than the high dose of 50 mg/kg p.o. of free curcumin by reducing levels of Aβ1-42-induced proinflammatory circulating cytokines, such as TNF-α, IL-6, IL-1β and IFN-γ, in the prefrontal cortex, hippocampus and serum of aged mice [215]. In addition, Sadegh et al. documented the low doses (4 mg/kg) of curcumin-nanostructured lipid carriers passing through the BBB and targeting oxidative stress markers (ADP/ATP ratio, lipid peroxidation and ROS formation) in the hippocampal tissue, resulting in enhanced learning and memory performance in vitro and in AD rats [216,217]. Moreover, curcumin loaded with chitosan and bovine serum albumin nanoparticles at a dosage of 100 μg/mL effectively increased the drug penetration across the BBB, promoted the phagocytosis of the Aβ peptide and further supported the activation of microglia by repressing TLR4-MAPK/NF-κB signaling and M1 macrophage polarization in vitro [218]. In particular, selenium nanoparticles raised the number of beneficial bacteria including *Bifidobacterium*, *Dubosiella*, *Desulfovibrio* and *Gordonibacter* to inhibit the Aβ aggregate-induced neurotoxicity, downregulating the expression of NLRP3 inflammasome and the inflammatory cytokine secretion nitrite (NO), interleukin-6 (IL-6), IL-1β and TNF-α, leading to neuroinflammation and death in vitro and in vivo models of AD [211]. A recent clinical study evaluated the effects of daily oral use of curcumin dispersed with colloidal nanoparticles (named Theracurmin^®^), a highly bioavailable form of curcumin, on memory performance. This study demonstrated that Theracurmin^®^ containing 90 mg of curcumin taken twice daily improved both the memory and attention in non-demented adults, and this was strongly associated with reductions in amyloid plaque and tau accumulation in the hypothalamus and amygdala after 18 months of treatment without any toxic effects [219].

### 5.2. Polyphenol-Nanoparticles for Gut Health

In the intestine, oral treatment (0.2%) with curcumin nanoparticles through the inhibition of NF-κB activation repressed mucosal inflammation and increased butyrate-producing bacteria, *Clostridium* cluster IV and XIVa, with an elevation of fecal butyrate levels and an expansion of regulatory T cells (Tregs) in the colonic mucosa in experimental animal models of IBD [220]. Notably, the prebiotic effects of oral polyphenol nanoparticles in modulating gut microbiota and brain function relieving anxiety- and depression-like behaviors and cognitive impairment in a mouse model of acute colitis have been also reported [218]. Specifically, an oral treatment (dose of 1 mg mL^−1^, 200 μL) of polyphenol armored with TNFα–small interfering RNA and gallic acid-mediated graphene quantum dot (GAGQD)-encapsulated bovine serum albumin nanoparticle, with a chitosan and tannin acid multilayer, prolonged the residence time in the colon and regulated gut microbial homeostasis by increasing *Lactobacillus* and inhibiting the expression of GABA receptors via gut–brain axis crosstalk, consequently alleviating neuroinflammation and improving the mood and cognitive recovery of IBD model mice [221]. Furthermore, it has been shown by Li et al. that an oral treatment of TGN-Res@SeNPs nanocomposites effectively improved AD by inhibiting Aβ aggregation and deposition in the hippocampus, decreasing ROS and enhancing antioxidant enzyme activities, leading to a downregulation of the neuroinflammatory cascade, particularly the NFκB/MAPK/Akt signaling pathway, and ultimately alleviating gut microbiota disorders by decreasing proinflammatory gut pathogens such as *Alistipes*, *Helicobacter*, *Rikenella*, *Desulfovibrio* and *Faecalibaculum* in cells and rodents [222]. Interestingly, the health effects of selenium nanoparticles coated with dihydromyricetin (DMY), a natural polyphenol extracted from grapevine tea (Tg-CS/DMY@SeNPs), in modulating gut microbiota diversity have been proven. Despite numerous preclinical studies, unfortunately, clinical trials investigating the potential effects of polyphenol nanoparticle delivery systems in the prevention and treatment of gut disorders are lacking [223,224]. A recent clinical study demonstrated that a treatment with Theracurmin^®^ (360 mg/day) for 3 months was effective and safe in patients with Crohn’s disease, and is capable of ameliorating symptoms [225]. Finally, a placebo-controlled trial revealed that an oral dose of 500 mg/tablet of curcumin–phospholipid supplementation (Meriva^®^) taken twice daily decreased plasma pro-inflammatory cytokines, such as monocyte chemoattractant protein-1 (MCP-1), IFN-γ and IL-4, and lipid peroxidation by modulating gut microbiota composition through a reduction in the bacterial community of *Enterobacter* and *Escherichia-Shigella*, and a significant increase in the relative abundance of *Lachnoclostridium*, *Lactobacillaceae* and *Prevotellaceae* after 6 months in subjects with chronic kidney disease [224]. Taken together, the data indicate that polyphenol-based nanoparticles could represent natural candidates for innovative therapeutic purposes in gastrointestinal and brain disorders as they are more bioavailable and stable in the intestine, and are capable of modulating the gut microbiota and crossing the BBB with potential pharmacological effects at lower doses than polyphenols alone.

## 6. The Probiotics for Gut and Brain Health

Gut–brain crosstalk influences neural, endocrine and immune pathways. Probiotics activating signaling molecules involved in the cellular stress response could represent an innovative strategy to counteract oxidative stress in gut–brain axis disorders. Recent evidence has documented that changes in the microbiota due to probiotic consumption can modify resilience to stress in vitro, in vivo and in humans [71,226,227,228]. 

### 6.1. Neuropsychiatric Disorders

Emerging evidence suggests the potential therapeutic benefits of psychobiotics acting through the gut–brain axis to affect neuronal development, function and behavior, representing a novel approach for mental health in neuropsychiatric disorders including anxiety and depression, ASD and schizophrenia [229]. In line with this, Hao et al. demonstrated that *Faecalibacterium prausnitzii* (ATCC 27766) exhibited psychobiotic effects by increasing cecal SCFAs and plasma IL-10 levels, and decreasing corticosterone and IL-6 levels in rats with anxiety and depression-like behaviors caused by chronic unpredictable mild stress [230]. Furthermore, *Bifidobacterium breve CCFM1025* significantly inhibited depression- and anxiety-like behaviors by mitigating HPA axis hyperactivity and inflammation, upregulating BDNF levels and downregulating c-Fos levels, raising the serotonergic system in the gut and brain of chronic stressed mice after 5 weeks [231].

#### 6.1.1. Depression and Anxiety

Recent evidence reported that *Lactobacillus gasseri* NK109 alleviated colitis and gut dysbiosis, leading to psychiatric disorders such as depression and memory deficits induced by exposure to *Escherichia coli* K1 by inhibiting neuroinflammatory NF-κB signaling and IL-1β expression and increasing BDNF levels in the hippocampus of mice via the gut–vagus nerve–brain axis [232]. Similarly, the oral administration of *Lactobacillus reuteri* NK33 in synergistic combination with *Bifidobacterium adolescentis* NK98 at a dosage of 1 × 10^9^ colony forming unit (CFU)/day significantly inhibited NF-κB activation, IL-6 expression and LPS levels, and induced hippocampal BDNF expression and CREB phosphorylation, alleviating anxiety and depression symptoms by suppressing gut dysbiosis through the inhibition of Proteobacteria in cells and in rodents [233]. In addition, the oral administration of *Lactococcus lactis* (1 × 10^9^ CFU mL^−1^) attenuated anxiety and depressive-like behaviors in response to chronic unpredictable mild stress with the improvement of 5-hydroxytryptamine (5-HT) metabolism in serum and colon via the modulation of gut microbiome composition after 5 weeks in vivo [234]. Numerous clinical studies highlighted the protective and therapeutic role of neuroactive probiotics in reducing the inflammatory cascade and kynurenine levels and increasing BDNF expression via a functional crosstalk between the gut–brain axis in major depressive disorders [235,236]. Indeed, the SANGUT study conducted in 120 patients with depression for 12 weeks demonstrated that a gluten-free diet and the synergistic combination of psychobiotics, particularly *Lactobacillus helveticus* and *Bifidobacterium longum* at a dosage of 3  ×  10^9^ CFU, inhibited the immune–inflammatory cascade and improved both psychiatric symptoms and gut barrier integrity [237]. In addition, a recent randomized, double-blind, placebo-controlled study demonstrated in 63 participants with chronic stress that the intake of *Lactiplantibacillus plantarum HEAL9* significantly reduced the plasma concentrations of two pro-inflammatory markers (soluble fractalkine and CD163) in subjects exposed to acute stress compared to placebo after four weeks [238]. Interestingly, gut inflammation induced by pathogenic bacteria and their products can directly reach the brain via the systemic circulation, triggering neuroinflammation with the activation of a cytokine cascade and the release of TNF-α, IL-6 and IL-1β, which in turn influenced various neuronal processes and promote depressive-like behaviors along the gut–vagus nerve–brain axis. Notably, gut microbiota inflammation triggers an alteration of tryptophan–kynurenine metabolism and the formation of neurotoxic quinolinic acid in the brain, resulting in mental health problems including depressive and schizophrenia symptoms [186]. Accordingly, a double-blind, randomized, placebo-controlled study conducted on 60 patients with depressive and anxiety symptoms showed that the probiotic bacteria *Lactobacillus Plantarum 299v* decreased plasma kynurenine levels, improving cognitive performance and function compared to the placebo [235]. Moreover, depressed patients had an altered fecal microbiota composition, with increased abundance of *Enterobacteriaceae* and *Alistipes* and reduced levels of *Faecalibacterium* compared to controls [27]. In addition, a randomized, double-blind, placebo-controlled clinical trial of 40 participants with a diagnosis of major depressive disorder (MDD) showed that probiotic capsules containing strains of *Lactobacillus acidophilus* (2 × 10^9^ CFU/g), *Lactobacillus casei* (2 × 10^9^ CFU/g) and *Bifidobacterium bifidum* (2 × 10^9^ CFU/g) increased antioxidant biomarkers including plasma GSH levels and decreased serum high-sensitivity C-reactive protein (hs-CRP) and insulin levels compared to the placebo after 8 weeks [239]. Lastly, a recent randomized placebo-controlled trial of 82 patients with depression documented that the probiotic OMNi-BiOTiC^®^ Stress Repair plus biotin with at least 7.5 billion organisms per 1 portion (3 g) increased beneficial bacteria including *Ruminococcus gauvreauii* and *Coprococcus* in the gut and upregulated vitamin B6/B7 metabolism, improving psychiatric symptoms compared to the placebo after 28 days of treatment [240].

#### 6.1.2. Autism

Importantly, a randomized controlled study demonstrated in children with ASD, anxiety and gastrointestinal symptoms that a probiotic formulation (Vivomixx^®^) containing 450 billion lyophilized bacterial cells belonging to *Lactobacillus* and *Bifidobacterium* taken twice a day exerted health effects not only in reducing gut dysbiosis but also in improving language and cognitive functions, as well as neurotransmission and connectivity attenuating inflammatory markers (TNF-α, IL-6) plasminogen activator inhibitor-1 (PAI-1) and chemical pollutants (phthalates) in plasma and urine after 6 months [241]. Another randomized, double-blinded, placebo-controlled pilot trial revealed that a daily intake of a *Lactobacillus plantarum PS128* probiotic (6 × 10^10^ CFU) in combination with oxytocin reduced serum inflammatory markers such as IL-1 β with positive socio-behavioral symptoms related to the increase of *Eubacterium hallii* in ASD patients after 28 weeks [242]. 

### 6.2. Neurodegenerative Disorders

Recently, probiotic supplements have been considered to improve cognitive function via the gut–brain axis in major neurodegenerative disorders including AD and PD [39,243]. Preclinical evidence demonstrated that a new formulation of lactic acid bacteria and *bifidobacteria* (SLAB51) at a dosage of 200 bn bacteria/Kg/day improved the cognitive function and plasma concentration of gut hormones such as ghrelin, leptin, GLP1 and GIP, as well as reduced pro-inflammatory cytokines, by modulating the gut microbiota composition with an increase in *Bifidobacterium* spp. and a reduction in *Campylobacterales* in 3xTg-AD mice after 24 weeks [244]. The same authors also observed that probiotic SLAB51 showed antioxidant effects, upregulating the Sirt1 pathway, which in turn reduced ROS production and preserved brain redox homeostasis in 3xTg-AD mice after 16 weeks [245]. Furthermore, Ma et al. demonstrated that a synergistic treatment with *Lactobacillus mucosae* NK41 and *Bifidobacterium longum* NK46 promoted anti-inflammatory effects by inhibiting LPS levels, NF-κB activation and TNF-α expression, resulting in an increase in BDNF expression, modulating the gut microbiota composition by enhancing the population of Bacteroides including *Odoribactericeae* and reducing Firmicutes and Proteobacteria phyla, as well as suppressing Aβ amyloid accumulation in the hippocampus of 5xFAD mice [246]. Numerous clinical trials reported that a daily intake of probiotics reduces oxidative stress, gut dysbiosis, neuroinflammatory cytokines and cognitive impairment in patients with AD and PD [247,248,249]. In congruence with this, a recent study observed that probiotic supplementation with kefir at the minimum dose of 2 ml/kg for 90 days influenced CNS function via the gut–brain axis, impacting positively on brain function (ameliorating memory, language, visual-spatial function, executive functions, conceptualization and abstraction abilities), systemic pro-inflammatory cytokines (downregulating IL-1, IL-6, IL-8, IL-12 and TNF-α), systemic antioxidant enzymes (upregulating GSH, GPx and SOD) and DNA repair/apoptosis (enhancing PARP-1 and p53 expression) in elderly patients with AD [247]. Recent randomized clinical controlled studies have demonstrated that the probiotic formulation containing *Lactobacillus acidophilus*, *Bifidobacterium bifidum* and *Bifidobacterium longum* (2 × 10^9^ CFU/day each) co-supplemented with selenium (200 μg/day) significantly improved cognitive function in AD patients [248] and the movement in PD patients [249] via a reduction in C-reactive protein and malondialdehyde, and an enhancement of antioxidant GSH after 12 weeks. In addition, a randomized study performed by Aljumaah et al. demonstrated that *Lactobacillus rhamnosus* GG reduced the relative abundances of *Prevotella ruminicola* and *Bacteroides thetaiotaomicron*, improving cognitive dysfunction in patients with MCI and promoting healthy brain aging in an elderly population [250]. Another recent randomized, double-blind placebo-controlled study conducted on 130 patients with MCI observed that a daily intake of *Bifidobacterium breve* MCC1274 (2 × 10^10^ CFU) for 24 weeks changed the gut microbiota composition by preventing the brain atrophy progression that ultimately improved the cognitive function in older patients with MCI [251]. Taken together, data from the emerging literature highlight that the gut microbiota and brain communicate with each other via the gut–brain axis, and this bidirectional crosstalk implicates that bacteria-derived metabolites can have a positive or negative impact on the CNS, leading to brain health and/or the onset of neurodegenerative disorders. Therefore, a proper dose of probiotics could represent great promise for the treatment of dementia in humans, as they interfere with a range of metabolic and antioxidant signaling pathways involved in the maintenance of cellular resilience homeostasis, which is of crucial importance for neural cells’ survival and the quality complex of human life. 

## 7. The Vagus Nerve a “Neurometabolic Sensor” of Gut–Brain Axis

The gut–brain axis involves the autonomic nervous system (ANS), including the parasympathetic nervous system, that is, the vagus nerve (VN), originating from the cranial parasympathetic nucleus and innervates numerous structures and organs such as the heart and gastrointestinal tract. The VN is the longest nerve of the organism and a key neural pathway between the gut and the brain, containing 80% and 20% of afferent and efferent fibers, respectively [252]. It plays an essential role in interoceptive awareness. Notably, the VN acts as a “neurometabolic sensor” because it is able to sense the microbiota metabolites through its afferent fibers to transmit this information from the gut to the brain, where it is then integrated in the central autonomic network to generate an adequate or maladaptive response. The latter could perpetuate a pathological state of the digestive tract or could favor psychiatric and neurodegenerative disorders, as well as gastrointestinal diseases [253,254]. Under physiological conditions, the VN is involved in gut and brain homeostasis because it senses the “milieu intérieur” of the gut through the interaction of nutrients and/or gut peptides (i.e., cholecystokinin, GLP-1, leptin and serotonin), with vagal afferents boosting the responses to maintain neurological and gastrointestinal health status [255]. The VN possesses anti-inflammatory properties by regulating gut dysbiosis due to peripheral cytokines release through its vagal afferent fibers and the activation of the HPA axis, which in turn stimulates the secretion of cortisol by the adrenal glands, whereas through vagal efferent fibers it induces a vagovagal reflex called the cholinergic anti-inflammatory pathway [256]. The cholinergic anti-inflammatory pathway is mediated by enteric neurons that innervate intestinal lamina propria to stimulate the release of acetylcholine (ACh) at the synaptic junction, which, via binding to the α-7-nicotinic ACh receptors (α7nAChR) of macrophages, ultimately inhibits the release of TNFα, IL-1β and IL-6 [256] (Figure 2). Perturbations of vagal homeostasis due to oxidative stress trigger a neuropathological cascade with the release of pro-inflammatory cytokines and glucocorticoids that act on vagal receptors and the microbiota, leading to dysbiosis and the onset of gastrointestinal disorders, ultimately promoting brain damage [257].

The stimulation of vagal afferent fibers in the gut increases the transmission of the monoaminergic brain system, especially norepinephrine, serotonin and dopamine, and recently it has emerged as an interesting nondrug therapy not only for the treatment of gastrointestinal and neurodegenerative disorders, such as IBD and PD, but also of major psychiatric conditions, such as depression and anxiety disorders, in which gut–brain crosstalk is dysfunctional [257,258]. More recently, effective natural therapeutic options, in particular, dietary supplementation with polyphenols and/or probiotics targeting antioxidant pathways to enhance the vagal tone and inhibit cytokine production in both neurological and gut disorders via VN modulation, have also been documented [259,260]. For this reason, polyphenols alone and/or in synergy with probiotics are important nondrug therapies to promote gut and brain resilience to pathological challenges (i.e., stress, inflammation and gut dysbiosis). In line with this, preclinical and clinical studies documented the healthy effects of probiotics in repressing neurological and gut disorders via VN stimulation [199,261,262,263]. Accordingly, Bravo et al. reported that *Lactobacillus rhamnosus* (JB-1) represses stress-induced corticosterone and anxiety- and depression-related behavior in mice through VN stimulation [199]. In addition, it has been also reported that *Lactobacillus reuteri* (~1 × 10^8^ organisms/day) reversed the social behavioral deficits in the gut in a VN-dependent manner via increasing the oxytocin levels in the brain and specifically in the ventral tegmental area of dopaminergic neurons, ultimately promoting synaptic plasticity between enteroendocrine cells and vagal afferents in mouse models of ASD [262]. Human studies, especially a double-blind randomized clinical trial on 60 women affected by normal weight obese and obesity, demonstrated that an intake for 3 weeks of a psychobiotics oral suspension containing strains of *Streptococcus thermophilus*, *Streptococcus thermophiles*, *Bifdobacterium animalis* subsp. *Lactis*, *Bifdobacterium bifdum*, *Lactococcus lactis* subsp. *Lactis*, *Lactobacillus delbrueckii* spp. *Bulgaricus*, *Lactobacillus acidophilus*, *Lactobacillus plantarum* and *Lactobacillus reuteri* at a dose of 3 g/day modulated the body composition, microbial contamination, psychopathological scores and eating behavior via the VN in pre-obese–obese women [263]. Intriguingly, Takada and colleagues demonstrated that a daily administration of milk fermented with *Lactobacillus casei* strain Shirota (1.0 × 10^9^ CFU/mL) for 8 weeks markedly repressed salivary cortisol levels, preventing the onset of physical symptoms in academically stressed students compared to the placebo group [261]. In support of this, the same authors also observed that an intragastric administration of *Lactobacillus casei* strain Shirota (2 × 10^10^ CFU/mL) stimulated gastric vagal afferent activity and sent sensory signals through the VN to the nucleus tractus solitarius (NTS) of the brain, thereby modulating HPA axis reactivity and subsequently suppressing the stress-initiated cortisol response dose-dependently [261]. Likewise, compelling evidence documented that a daily intake of polyphenols, such as resveratrol and polyunsaturated fatty acids (PUFAs), modulates the gut microbiota via the endocannabinoid pathway and light, odor and taste receptors, which, in turn, transfer their messages to the other organs, in particular to the brain via the VN [264]. The protective mechanism of resveratrol on the intestinal microbiota is mediated by the upregulation of the Sirt1 pathway, which increases insulin sensitivity and reduces hepatic glucose production through the activation of the hypothalamic K_ATP_ channel and the innervation of the hepatic VN [265]. Importantly, polyphenols, in particular flavones and flavone metabolites, are able to modulate neural transmission. In this regard, Ishii et al. observed that a single oral dose of 50 mg/kL of flavan-3-ols, a catechin and procyanidin fraction, crosses nerves and affects adipose tissue through sympathetic nerve activation and by upregulating thermogenic transcription factors, such as eroxisome proliferator-activated receptor γ coactivator (PGC)-1α, PR domain-containing PRDM16 and mitochondrial uncoupling protein 1 (UCP-1), in mouse adipose tissues [266]. The active flavonoid isoliquiritigenin (10 μM) contained in *Glycyrrhiza glabra* L. root, Kaempferol-3-rhamnoside and rosmarinic acid exerted neuroprotection, activating GABA receptors and decreasing intracellular Ca^2+^ and glutamate release into cerebrocortical terminals from synaptic vesicles in murine models [267,268,269]. In addition, a dose of 25 or 50 mg/kg of epigallocatechin gallate remarkably attenuated the neuronal expression of NADPH-d/nNOS and cell death in the motor neurons following peripheral nerve injury [270]. Equally importantly, genistein isoflavone, a tyrosine kinase inhibitor, reduced the Ca^2+^ influx through T-type Ca_V_3.3 voltage-gated ion channels, affecting nerve activation in a concentration-dependent manner in vitro and in silico [271]. Furthermore, it has been reported that a daily supplementation with matured hop-derived bitter acids (35 mg/day) found in beer enhanced the hippocampal memory and prefrontal cortex-related to cognitive function via VN stimulation in rodent models of neurodegeneration [272]. Recently, Skiba et al. evaluated the pharmacometabolic effects of pteryxin (20 μM), a natural coumarin compound, demonstrating that this substance purposely augmented and restored whole-body acetylcholine, choline and serotonin levels in zebrafish larvae mediated by VN stimulation. Therefore, pteryxin could be considered an adjunctive therapeutic approach to treat refractory epilepsy [273]. Interestingly, it has been also demonstrated that ayurvedic medicine, particularly *centella asiatica* extract (200 mg), alleviated colitis by inhibiting inflammatory cell infiltration with reduced myeloperoxidase activity in the colon and upregulated the expression of tight junction protein (ZO-1, claudin-1 and E-cadherin) to enhance the intestinal mucosa permeability and the abundance of beneficial bacteria, such as the *Firmicutes* phylum, and reduce the abundance of the *Proteobacteria* phylum in order to restore intestinal motility and promote c-Kit expression in the colon and 5-HT in the brain [274]. Finally, a very recent randomized, double-blind, placebo-controlled cross-over trial performed on 11 healthy subjects observed that the consumption of 500 mg of UP360 containing *Aloe vera*, *Poria cocos* mushroom and *Rosmarinus officinalis* extract increased chemokine levels through the activation of innate immune cell markers (i.e., NKT cells, monocytes, CD8^+^ T cells and γδT cells), which was followed by an increase in the antioxidant pathways (i.e., superoxide dismutase and catalase), leading to a significant reduction in TNF-α levels via vagal communication [275]. Overall, the therapeutic effects of psychobiotics and/or natural compounds on intestinal inflammation and brain damage are likely due to the activation of the Nrf2-dependent antioxidant pathways that inhibit gut-mediated cytokine release via VN stimulation.

## 8. Ferroptosis and Nrf2 Signaling in Gut–Brain Axis: Relevance to Natural Therapy

Ferroptosis is a newly iron- and lipid peroxidation-dependent cell death cascade first described by Dixon and colleagues in 2012 [276]. It is caused by the redox state depletion of the intracellular antioxidant microenvironment, which is tightly controlled by glutathione (GSH) and glutathione peroxidase 4 (GPX4) and is responsible for regulating ROS levels via the Nrf2 pathway. Therefore, the depletion and inhibition of GSH antioxidant levels inactivates and represses the decomposition of toxic lipid hydroperoxides into lipid alcohols to initiate ferroptosis. Ferroptosis is thus promoted by the suppression of cysteine uptake, reduced GSH levels or inactivation of the lipid repair enzyme GPX4. Emerging evidence indicates that lipid peroxidation and GSH metabolism dysfunction trigger ferroptosis by promoting neuronal loss and CNS-associated damage in a range of neurodegenerative disorders, as well as affecting gut microbiota homeostasis, leading to the pathogenesis or progression of gastrointestinal diseases [277]. Nrf2 is known to regulate ferroptosis in multiple pathways and the mechanism of action in which natural compounds target Nrf2 to inhibit ferroptosis is attracting the attention of many researchers. It is equally noteworthy that the upregulation of Nrf2 and its target *vitagenes* (HO-1, NQO-1 and Trx) activated by dietary polyphenols, such as flavonoids, contributes not only to neuroprotection, delaying brain degeneration [278], but is also highly linked to gut microbiota homeostasis during ferroptosis [279,280]. Consistent with this, recent preclinical evidence elucidated the underlying molecular mechanism of quercetin, a natural flavonoid, on gastrointestinal inflammation and revealed that at a concentration of 50 mg/kg it reduced dietary mycotoxins-induced intestinal ferroptosis by attenuating the inflammatory TLR4/NF-κB signaling pathway in mice [281]. In addition, quercetin reduced lipid peroxidation (e.g., downregulating 4-HNE and upregulating the GSH/GSSG ratio) and lipid accumulation, inhibiting hepatic ferroptosis in vitro and in vivo [282]. In the brain, low doses of quercetin have shown significant neuroprotective effects by attenuating cognitive impairment and ameliorating motor behavioral disorders via the inhibition of ferroptosis and the activation of the Nrf2 pathway in vitro and in vivo models of AD and PD [283,284]. In addition, curcumin has been reported as a novel ferroptosis inhibitor due to its protective effects as an iron chelator, thus preventing GSH depletion, GPX4 inactivation and lipid peroxidation in pancreatic cells [285], as well as 6-hydroxydopamine-induced nigral dopaminergic neuronal degeneration in PD rats [286] in a dose-dependent manner. Sulforaphane is known as an Nrf2 activator, a mechanism of action for anti-oxidative and anti-inflammatory activities and a key negative regulator of ferroptosis [287]. Notably, evidence indicated that sulforaphane inhibited ferroptosis, downregulating lactate dehydrogenase, Fe^2+^, malondialdehyde and acyl-CoA synthetase long-chain family member 4 (ACSL4), and upregulating Nrf2, GSH, glutathione peroxidase 4 (GPX4) and solute carrier family 7 member 11 (SLC7A11) in vitro and in vivo [288] (Figure 3).

Similarly, epigallocatechin-3-gallate (EGCG) protected against radiation-induced intestinal damage by scavenging ROS and inactivating ferroptosis via the upregulation of the Nrf2 pathway and its antioxidant proteins SLC7A11, HO-1 and GPX4 [289]. A recent study showed that EGCG exhibits therapeutic effects against the nonalcoholic steatohepatitis induced by a deficient diet in methionine and choline through shifting gut dysbiosis (i.e., *Oxalobacter*, *Oscillibacter*, *Coprococcus_1* and *Desulfovibrio* versus *Bacteroides*, *Bifidobacteria* and *Lactobucillus*), improving the hepatic injury, lipid accumulation, fibrosis and ferroptosis pathway after 4 weeks in animals [280]. Ginkgolide B, a terpenoid found in *Ginkgo biloba*, by targeting the Nrf2/GPX4 signaling pathway, has been shown to ameliorate AD-related cognitive impairment in senescence mice by decreasing the iron content in the brain, transferrin receptor 1 (TFR1) and nuclear receptor coactivator 4 (NCOA4) expression, and by increasing ferritin heavy chain (FTH1) expression [290]. Interestingly, salidroside, a polyphenol contained in *Rhodiola Rosea* L., effectively reduced intracellular Fe^2+^ levels, attenuating lipid peroxidation and mitochondrial damage and enhancing the expression of GPX4 and SLC7A11 by targeting Nrf2/HO1 signaling to block neuronal ferroptosis in hippocampal cells and Aβ_1-42_-induced AD mice [291]. In the same way, the probiotic formulation containing three strains of Lactobacilli (*Lacticaseibacillus rhamnosus* LR04 (DSM16605), *Lactiplantibacillus plantarum* LP14 (DSM33401) and *Lacticaseibacillus paracasei* (LPC09)) and two strains of Bifidobacteria (*Bifidobacterium breve* BR03 (DSM16604) and B632 (DSM24706)) completely restored the harmful effects of ferroptosis and gut inflammation due to dinitrobenzene sulfonic acid exposure in colon tissues of ex vivo models (organ-on-chip) [292]. Taken together, the abovementioned recent findings elucidate the crucial role of the Nrf2 pathway and *vitagenes* activated by natural mitigators (i.e., polyphenols or probiotics) as a rational line of therapy for ferroptosis-induced neurodegeneration and intestinal dysfunction.

## 9. Nutritional Therapeutic Interventions Using Innovative In Vitro Modeling

Recent research in stem cell biology has led to the successful three-dimensional culture of tissue in vitro, also known as “organoids”, or three-dimensional organ-like structures composed of functional, live cells that can self-renew and spatially organize. The ENS strongly influences mucosal immunity and epithelial function, and is currently suggested as an important contributor to IBD development and progression [293]. On the other hand, ENS dysfunction may lead to bidirectional consequences for the gut and the nervous system, as various neurotransmitters are released from ENS neurons and can in turn act on the intestinal epithelium (gut–brain axis), exacerbating peripheral inflammation and the development of neuropathologies [294]. Notably, the direct crosstalk between the gut epithelium and specific primary afferent fibers is conducted by enterochromaffin cells (EC), which are proposed as chemosensors of the intestinal epithelial that modulate vagal sensory neurons, bacteria products and metabolites, chemical irritants, inflammatory mediators and neurotransmitters from the gut directly to the nervous system [295]. In this light, intestinal and brain organoids provide an innovative in vitro platform to explore cellular communication and molecular mechanisms of host–microbe interaction and inter-organ crosstalk between the intestine and CNS in health and/or disease. To note, the discovery of novel natural therapeutic candidates, including polyphenols and/or probiotics tested on organoids, could be relevant for personalized nutritional medicine of gut and brain disorders, ultimately revolutionizing research in the fields of neuroscience and gastroenterology (Figure 4) [296]. Recent findings have found that ferulic acid, a polyphenolic compound, promotes the survival and differentiation of mouse intestine organoids, prevents inflammation and ensures gut health by protecting the intestinal epithelial barrier; thus, it could be considered as a potential preventive or alleviating component in the diet of IBD patients [297]. Furthermore, a study revealed that a low dose (0.1%) of mixture composed of functional amino acids (L-arginine, L-leucine, L-valine, L-isoleucine and L-cystine) and 100 ppm of a polyphenol-rich extract from grape seeds and skins regulates epithelial homeostasis and modulates the gut microbiota in vivo and in intestinal organoids [298]. In addition, another study by Tveter et al. demonstrated that the administration of proanthocyanidin-rich extract of grape polyphenols improves glucose metabolism, decreases pathogenic gut bacteria associated with nuclear transcription factor farnesoid X receptor (FXR) inhibition and downregulates ceramide synthesis genes (e.g., *Smpd3*, *Cers4* and *Sptlc2)* in the intestine of *db*/*db* mice after 4 weeks [299]. Interestingly, a recent research conducted by Elbadawi et al. showed that a low dose of 10 g/mL of a new herbal preparation termed STW 5-II and consisting of six synergistic medicinal extracts, i.e., *Iberis amara* L., *Glycyrrhiza glabra* L., *Chamomilla recutita* R., *Menthae piperitae* L., *Melissa officinalis* L. and *Carum carvi* L., reduced levels of corticotropin-releasing factor (CRF), mediating IL-6, IL-1β and TNFα expression and serotonin release, using mouse intestinal organoids as a model of IBS disorder dose-dependently [300]. Likewise, the application of intestinal organoids and the effectiveness of probiotics in maintaining gut epithelial regeneration and homeostasis against inflammation have been extensively documented [301,302,303]. In particular, Hou et al. showed the efficacy of *Lactobacillus reuteri* D8 (10^6^ CFU/g) in regenerating the intestinal barrier and activating the intestinal epithelial proliferation of organoids subjected to TNF-α damage. The authors observed that *Lactobacillus reuteri* D8 reduced TNF-α and stimulated intestinal organoids with lamina propria lymphocytes (LPLs) to secrete IL-22 via STAT3 pathway ex vivo and in vivo [301]. Furthermore, Wu and coworkers confirmed that *Lactobacillus reuteri* D8 protected the intestinal mucosal barrier’s integrity, repaired intestinal damage and reduced inflammation induced by TNF in intestinal organoids or C. *rodentium* infection in mice via the activation of the Wnt/β-catenin signaling pathway [302]. Intriguingly, other recent evidence showed postbiotic effects of a new probiotic strain of *Lactobacillus reuteri* DS0384 isolated from the feces of a healthy newborn, and found that it promotes intestinal epithelial maturation and protects the intestinal epithelium from IFNγ/TNFα-induced injury in human intestinal organoids and infant mice [303]. Furthermore, Engevik et al. demonstrated that *Lactobacillus reuteri* ATCC PTA 6475 metabolites, including ethanol, upregulate the serotonin transporter (SERT), restoring normal serotonin levels for maintaining intestinal homeostasis in mouse colonic organoids [304]. The same authors also investigated another probiotic, *Bifidobacterium dentium*, and showed that its metabolites, particularly acetate, stimulated enterochromaffin cells to secrete 5-hydroxytryptamine (5-TH) within the intestinal epithelium of human enteroid cells and adult mice via the gut–brain axis. Notably, the study revealed that *Bifidobacterium dentium* changed the expression of key intestinal serotonin receptors, particularly isoforms 2a (*Htr2a*) and 4 (*Htr4*), and the 5-HT transporter, a serotonin transporter (*Sert*), upregulating the expression of *Htr2a* in the hippocampus and normalizing anxiety like-behaviors in mice [305]. Most recently, an intriguing study showed the beneficial effects of the intestinal bacterium *Lactobacillus plantarum* in modulating the intestinal microbiota and improving gut barrier integrity by reducing inflammation-related pathways (TNF-alpha and NF-κB) and increasing levels of L-arginine in both hepatic and intestinal organoids and in mouse models of nonalcoholic steatohepatitis [306]. Concerning neurodegenerative disorders, recent advances using the human midbrain organoids documented that mutation in the *DNAJC6 gene* encoding HSP40 auxilin caused PD pathogenesis including autophagy defects, α-syn aggregation and dopaminergic neuron degeneration, contributing to juvenile-onset PD [307]. Lastly, a recent study conducted by Ji and collogues reported that exosomes from brain organoids relieved H_2_O_2_-induced oxidative stress and apoptosis in rat midbrain astrocytes by suppressing ROS overproduction, lipid peroxidation, mitochondrial dysfunction and the expression of pro-apoptotic genes, and they promoted the differentiation of human-induced pluripotent stem cells (iPSCs) into dopaminergic neurons via homeobox transcription factor 1 alpha (LMX1A) upregulation due to increased levels of neurotrophic factors including neurotrophin-4 (NT-4) and glial-cell-derived neurotrophic factor (GDNF) [308]. Taking together, to date, there are few studies in the literature that have investigated the therapeutic potential of a nutritional approach with polyphenols and/or probiotics using organoid models to prevent or inhibit gut and brain inflammation and related disorders in order to improve human health. It is hoped that in the future personalized nutritional medicine in this interesting field will become more promising and will be taken into consideration.

An altered gut–brain crosstalk due to an increase in pathogenic bacteria, such as *Streptococcoceae*, *Clostridiaceae*, *Enterobacteriaceae* and *Proteobacteriaceae*, as well as a reduction in beneficial bacteria, such as *Bifidobacteriaceae* and *Lactobacillaceae*, triggers gut dysbiosis and neuroinflammation, which in turn leads to enteric glia and microglia activation and, consequently, BBB and IEB dysfunctions, ultimately culminating in the development of gastrointestinal diseases including inflammatory bowel diseases (i.e., ulcerative colitis and Crohn’s disease), pancreatitis and colorectal cancer and CNS disorders (i.e., AD, PD, depression and anxiety, autism and schizophrenia). Personalized nutritional therapy through polyphenols and probiotics at low doses (hormesis) tested in innovative in vitro models, especially the brain and intestinal organoids, can prevent or inhibit an inflammatory cascade and restore the physiological composition of the gut microbiota and subsequent brain function in order to promote human health. ↑ increase, ↓ decrease.

## 10. Conclusions and Future Perspectives

In conclusion, both indirect and direct crosstalk between intestinal microbiota and the CNS along the gut–brain axis provides a rationale for non-invasive and affordable therapeutic innovations in brain disorders including neurodegenerative and psychiatric diseases. In this new way, hormetic nutrition through polyphenols and/or probiotics targeting the antioxidant Nrf2 pathway and stress resilient *vitagenes* to inhibit oxidative stress and inflammatory pathways, as well as ferroptosis, could represent an effective therapy to manipulate alterations in the gut microbiome leading to brain dysfunction in order to prevent or slow the onset of major cognitive disorders. Notably, hormetic nutrients can stimulate the vagus nerve as a means of directly modulating microbiota–brain interactions for therapeutic purposes to mitigate or reverse the pathophysiological process, restoring gut and brain homeostasis, as reported by extensive preclinical and clinical studies. Interestingly, emerging research highlighted that gut microbiome composition and function can predict healthy aging and longevity in humans. Indeed, a positive correlation was attributed to the phylum of *Firmicutes* and a negative association was attributed to the abundance of *Bacteroides* [309]. Notably, the abundance of pathogenic bacteria involves a shift towards a pro-inflammatory state in the gut, yielding increased gut permeability and subsequent triggering of a peripheral inflammatory response, which in turn impacts vagal sensory neurons, culminating in neuroinflammation and impaired cognitive function. Therefore, a deeper understanding of the composition, function and expression of gut microbiota bacteria, as well as their modulation through low doses (hormesis) of probiotics in synergistic combination with polyphenols, and especially with the more bioavailable polyphenol nanoparticles, can restore intestinal homeostasis and promote brain health (allostasis/adaptive response) in both young and elderly populations. Lastly, novel and sophisticated in vitro modeling through the use of gut and brain organoids to explore the relationship between the intestinal microenvironment, host–microbes interaction and inter-organ crosstalk, as well as the underlying mechanism of action of polyphenols/probiotics targeting the antioxidant Nrf2 pathway, could shed more light on personalized nutritional therapies to prevent or attenuate intestinal dysbiosis and peripheral inflammation/neuroinflammation leading to the degeneration of sensory neurons and cognitive impairment, and could ultimately predict positive outcomes in gut–brain axis disorders.

## Figures and Tables

**Figure 1 antioxidants-13-00484-f001:**
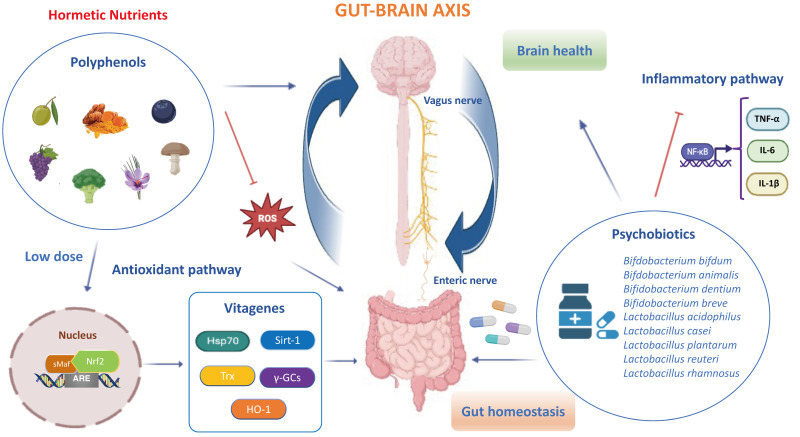
Hormetic nutrition modulates redox stress resilience *vitagenes* via gut–brain axis.

**Figure 2 antioxidants-13-00484-f002:**
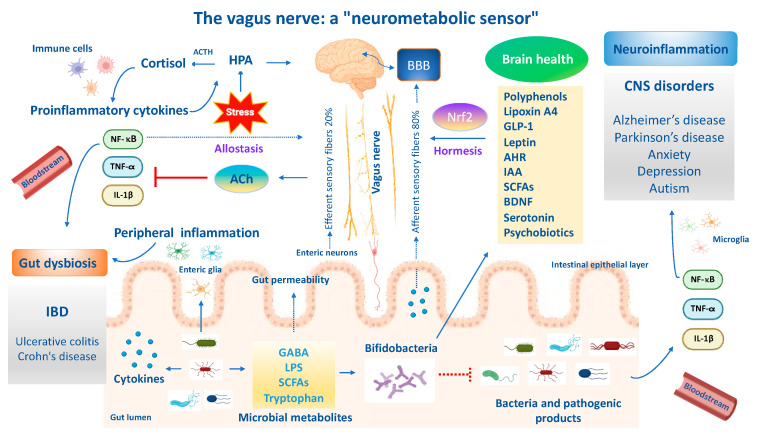
Schematic overview of the vagus nerve as “neurometabolic sensor” of the gut–brain axis. The vagus nerve is a “neurometabolic sensor” of the gut–brain axis, since it senses microbiota metabolites, including SCFAs, LPS, tryptophan and GABA, generated by pathogenic bacteria, as well as those produced by beneficial bacteria such as *Bifidobacteria* through its afferent fibers, to transfer this gut information to the brain, where it is integrated into the central autonomic network and then generates an adapted or inappropriate response. The abundance of pathogenic bacteria implicates a shift towards a pro-inflammatory state in the gut, yielding increased gut permeability and the subsequent triggering of a peripheral inflammatory response, which in turn impacts vagal sensory neurons through its afferent fibers, culminating in neuroinflammation and impaired cognitive function. Elevated inflammatory cytokines in the bloodstream and stress activate the HPA axis through ACTH, leading to cortisol secretion which affects the immune cells and contributes to cytokines release. On the other hand, the efferent sensory fibers activate the cholinergic anti-inflammatory pathway that stimulates the release of acetylcholine (ACh) to inhibit the secretion of proinflammatory cytokines such as TNFα, Il-1β and Il-6. At the appropriate dose, hormetic nutrients, especially polyphenols in synergy with psychobiotics, as well as *Bifidobacteria* produced-metabolites such as lipoxin A4, GLP-1, leptin, SCFAs, BDNF, IAA, AHR and serotonin targeting the Nrf2 pathway, can directly stimulate the vagus nerve through its afferent sensory fibers to attenuate or reverse the pathophysiological process, converging in the onset of IBD and central nervous system disorders in order to restore gut and brain homeostasis to environmental challenges (hormesis/allostasis response). IBD: inflammatory bowel diseases, CNS: central nervous system, AHR: aryl hydrocarbon receptor, IAA: indole-3-acetic acid, LPS: lipopolysaccharide, HPA axis: hypotalamic pituitary-adrenal axis, ACTH: adrenocorticotropic hormone, SCFA: short-chain fatty acids, GABA: γ-aminobutyric acid.

**Figure 3 antioxidants-13-00484-f003:**
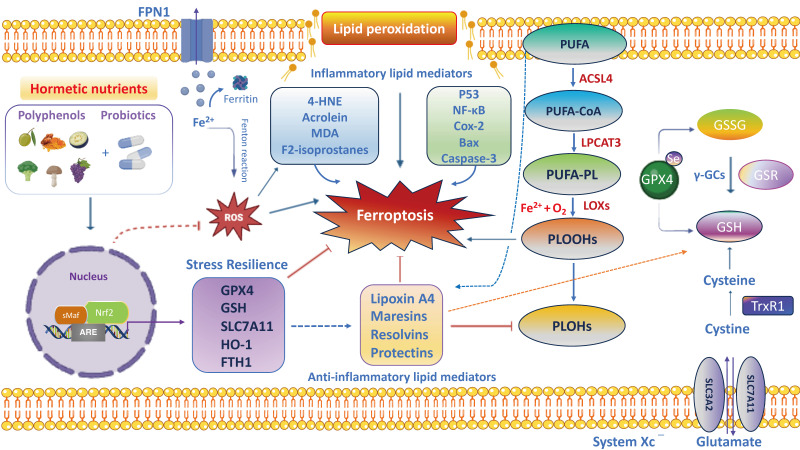
Potential mechanism of action of nutrients in preventing ferroptosis via Nrf2 pathway. Hormetic nutrients, such as polyphenols, in synergy with probiotics upregulate the Nrf2 antioxidant pathway and intracellular resilience scavengers including GPX4, GSH, SLC7A11, HO-1, ferritin heavy chain 1 (FTH1), which in turn activates anti-inflammatory lipid mediators such as lipoxin A4, maresins, resolvins and protectins generated by polyunsaturated-fatty-acid (PUFA) to block ferroptosis and lipid peroxidation triggered from pro-inflammatory lipid mediators such as 4-HNE, acrolein, COX-2 and MDA; F2-isoprostanes; inflammatory cytokines including NF-κB; and apoptotic mediators such as P53, Bax and Caspase-3 [277,280,289]. In particular, GPX4 is able to oxidize glutathione (GSH) to oxidized glutathione (GSSG) to reduce lipid peroxides and restore cellular redox homeostasis. GSH-reductase (GSR) reproduced GSH from GSSG, which was also synthesized by γ-GCs. Cystine enters cells through a system Xc- transporter and is reduced to cysteine via the thioredoxin reductase 1 (TrxR1) to synthesize GSH. The inhibition of cystine input blocks the synthesis of GSH in cells and promotes ferroptosis. PUFA activated by acyl-CoA synthetase long-chain family member 4 (ACSL4) starts lipid peroxidation and converts to PUFA acyl-CoA to generate phospholipid (PUFA-PL) by lysopho-sphatidylcholine acyltransferase 3 (LPCAT3) in the lipid membrane, which reacts with molecular oxygen through the Fenton reaction to generate peroxide radicals that promote the dehydrogenation of PUFA to toxic phospholipid hydroperoxide (PLOOHs), which is mediated by lipoxygenases (LPXs) to drive lipid peroxidation and form the corresponding alcohol (PLOH) and lipid radicals. The storage of iron is mediated by ferroportin 1 (FPN1) and ferritin is responsible for the regulation of iron homeostasis. Lipoxin A4 and other anti-inflammatory lipid mediators activated by nutrients block the toxic lipid cascade via activation of the Nrf2 pathway and antioxidant systems of crucial importance for maintaining functional intracellular repair mechanisms such as GPX4 and GSH, thus preventing ferroptosis in both the gut and the brain.

**Figure 4 antioxidants-13-00484-f004:**
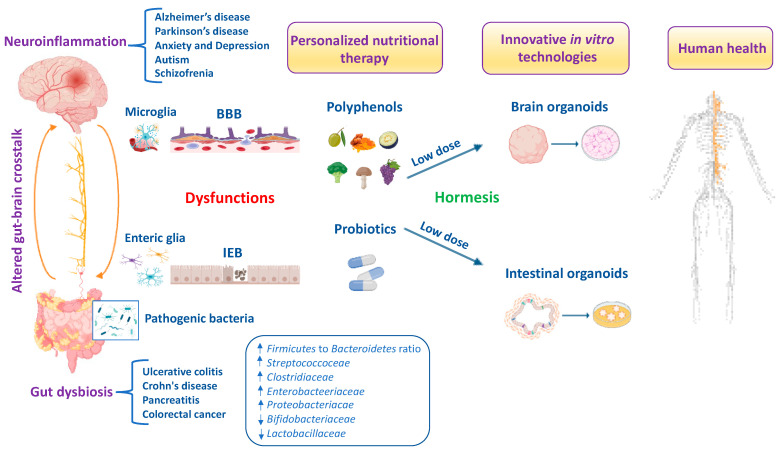
Personalized nutritional therapy using organoid models in gut–brain axis disorders.

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
