# Peer review of "Hormetic Nutrition and Redox Regulation in Gut–Brain Axis Disorders"

_antioxidants, 2024, doi:10.3390/antiox13040484_

Round 1

Reviewer 1 Report

My concerns are related to the following aspects:

1. Paragraph Structure and Formatting:

2. Figures and Illustrations:

3. Writing Clarity:

4. Missing Citations:

This manuscript addresses a fascinating and timely topic. Here are my concerns/suggestions, which would help the authors to improve the clarity, flow, and overall impact of their manuscript.

1. Paragraph Structure and Formatting:

For better readability, please consider recommending the authors break up lengthy paragraphs, especially in sections like:

·       Introduction (pages 1-2)

·       Polyphenol-nanoparticle delivery systems (pages 11-12)

·       Neuropsychiatric disorders & Neurodegenerative disorders (pages 12-15)

·       Vagus nerve (page 15)

·       Ferroptosis and Nrf2 signaling (pages 15-19)

·       Nutritional interventions (pages 15-19)

To split these lengthy paragraphs into smaller paragraphs based on logical ideas would improve writing flow.

2. Figures and Illustrations:

  • I would like to suggest adding brief descriptions or explanations to Figure legends for Figures 1 and 2.
  • I would like to recommend including additional schematics to illustrate:
    • The vagus nerve as a "neurometabolic sensor" in the gut-brain axis.
    • The mechanisms of Ferroptosis and Nrf2 signaling in the gut-brain axis. These visuals would significantly enhance reader comprehension.

3. Writing Clarity:

On page 1, the authors refer to "the second brain" or "the forgotten endocrine organ" without specifying what "it" is. Clarify that this refers to the gut microbiota, the complex ecosystem of microorganisms in the gastrointestinal tract.

Example References are as follows:

  • Clarke G, et al. (2014) [Minireview: Gut microbiota: the neglected endocrine organ. Mol Endocrinol. 28(8):1221-38]
  • Donati Zeppa S, et al. (2022) [Interventions on Gut Microbiota for Healthy Aging. Cells. 12(1):34]

4. Missing Citations:

It seems that citations are required for the following paragraphs/sentences on page 4:

  • "Moderate intake of hormetic nutrients...promote gut and brain healthy effects..."
  • "The possibility of adopting natural antioxidant therapies..."

To include citations would strengthen the evidence and credibility of these claims.

Please find that my annotations about the above comments in the attached file.

Author Response

Dear Reviewer the revised points are attached.

Reviewer 2 Report

This is a comprehensive review of the literature regarding hormetic nutrients and vitagenes for the management of oxidative stress, inflammation, and microbiota dysregulation, and to enhance overall health in humans. The 20-page manuscript is well written, thorough and, with 271 citations, has extensively cited the relevant and current literature. 

Line 85: add "of" - "thousands of metabolites"...

Line 507-508: "lymphocytes"

Lines 519-520: I believe curcumin would be loaded into the nanoparticles, not the other way around.

Line 524: Reference 168 has been retracted by the journal.

Author Response

Dear reviewer, your important observations have been revised in the text. Thank you very much.

Reviewer 3 Report

This paper discusses an interesting nutritional model that applies the concept of hormesis to explain the positive impact of the specific use of phyto-compounds and secondary metabolites of plants at the level of the antioxidant response in the human body and its protective action against reactive oxygen and nitrogen species, the increase in which, if not balanced, has been linked to the altered activity of the gut-brain axis and, consequently, to the onset of gastrointestinal illnesses and even neuropathology. The article is overall well presented, and the authors clearly explain their arguments and issues based on the hormesis hypothesis and the effect of low doses of plant defense molecules in preventing the above diseases.

There are some aspects that should be improved in the manuscript by the authors to give greater strength to their model.

1) First, they should include a paragraph in the manuscript describing the relationships between allostasis and hormesis , since these concepts are both linked to resilience, recovery and modulation of redox responses.

2) They should introduce the possible interaction between the Nrf2 signaling pathway, the gut microbiota and other paths linked to the stress response and resilience, such as other networks involved in the dysregulation of the hypothalamus-pituitary-adrenal axis, for instance by highlighting the role of genetic and epigenetic aspects, or the interactions with neurotransmitter/neuropeptide systems and including the metabolism of some amino acids as tryptophan, phenylalanine, tyrosine and the sulfur-containing ones; also, the interactions between the Nrf2 signaling pathway and the aryl hydrocarbon receptor one should be indicated. These points could be considered redundant or not relevant, but, on the other side, it is instead important to better convey the complexity of the topic.

3) On the basis of the concept of hormesis implying biphasic responses to phyto-constituents and their potential benefits, even low-dose substances displaying toxic properties could stimulate the activation of antioxidant or detoxification pathways. The manuscript rather deals with the use of molecules endowed themselves with antioxidant properties. The authors should mention this concern to better introduce the beneficial effects of the compounds they reported in the manuscript in respect to their nutritional hormesis model.

 4) The authors should consider the limitations of their nutritional model and include them in the manuscript. The authors should therefore propose future development of the research in the field.

Author Response

Dear Reviewer your important annotations has been revised in the text.

Round 2

Reviewer 1 Report

The authors have sufficiently addressed my concerns. As a result of their revision, the quality of the manuscript has been significantly improved.

Please make sure to do more proofreading and upload figures with high-resolutions. 

Reviewer 3 Report

The revised version of the manuscript is now acceptable for publication. All suggested points and issues have been now considered in the manuscript. The article is exhaustive and complete. I would anyway suggest to the authors, at their discretion, to still include possible limitations of the model or difficulties of investigation in accordance with it. In my opinion it would greater impact to the article, increasing the scientific interest on the topic. 

The revised version of the article is ok.